# High-throughput screening of human genetic variants by pooled prime editing

## Graphical abstract

## Authors

Michael Herger, Christina M. Kajba, Megan Buckley, Ana Cunha, Molly Strom, Gregory M. Findlay

## Correspondence

greg.findlay@crick.ac.uk

## In brief

Herger and Kajba et al. introduce a prime editing platform to screen genetic variants for functional effects in haploid human cells. Negative and positive selection screens identify loss-of-function variants in coding and non-coding regions of *SMARCB1* and *MLH1*, illustrating how this approach can accelerate the discovery of variants underlying disease.

## Highlights

- A prime editing platform is developed in HAP1 cells to assay human variants at scale

- Co-selection for edited cells and inclusion of surrogate targets enhance data quality

- Negative and positive selection screens reveal new LoF variants in *SMARCB1* and *MLH1*

- Potential to boost discovery of pathogenic variants in coding and non-coding regions

Herger et al., 2025, Cell Genomics 5, 100814
April 9, 2025 © 2025 The Author(s). Published by Elsevier Inc.

 CellPress

CellPress

## Article

# High-throughput screening of human genetic variants by pooled prime editing

Michael Herger,[1,3] Christina M. Kajba,[1,3] Megan Buckley,[1] Ana Cunha,[2] Molly Strom,[2] and Gregory M. Findlay[1,4,*]
[1]The Genome Function Laboratory, The Francis Crick Institute, London NW1 1AT, UK
[2]Viral Vector Core, Human Biology Facility, The Francis Crick Institute, London NW1 1AT, UK
[3]These authors contributed equally
[4]Lead contact
*Correspondence: greg.findlay@crick.ac.uk

## SUMMARY

Multiplexed assays of variant effect (MAVEs) enable scalable functional assessment of human genetic variants. However, established MAVEs are limited by exogenous expression of variants or constraints of genome editing. Here, we introduce a pooled prime editing (PE) platform to scalably assay variants in their endogenous context. We first improve efficiency of PE in HAP1 cells, defining optimal prime editing guide RNA (pegRNA) designs and establishing enrichment of edited cells via co-selection. We next demonstrate negative selection screening by testing over 7,500 pegRNAs targeting *SMARCB1* and observing depletion of efficiently installed loss-of-function (LoF) variants. We then screen for LoF variants in *MLH1* via 6-thioguanine selection, testing 65.3% of all possible SNVs in a 200-bp region including exon 10 and 362 non-coding variants from ClinVar spanning a 60-kb region. The platform's overall accuracy for discriminating pathogenic variants indicates that it will be highly valuable for identifying new variants underlying diverse human phenotypes across large genomic regions.

## INTRODUCTION

Experiments to determine how genetic variants alter function can inform mechanisms, provide evidence for causal associations underlying human disease, and improve computational tools for variant-effect prediction.[1–3] Attempts to leverage human genetic data for both biological discovery and precision medicine, however, have been hindered by a shortage of functional evidence. Despite improving performance of computational predictors,[4–6] rare variants are still highly challenging to interpret, both in coding and non-coding regions. Most variants observed in humans have never been assayed for functional effects, and hundreds to thousands of variants of uncertain significance (VUSs) have been reported in each of many clinically actionable genes.[7,8]

Multiplexed assays of variant effect (MAVEs) have emerged as a means of providing functional evidence of variant effects at scale.[9,10] While numerous MAVEs have demonstrated high predictive power for identifying disease-associated variants,[11–13] technologies using genome editing to install variants have proven particularly accurate at detecting loss-of-function (LoF) variants in disease genes,[14–16] owing to advantages conferred by assaying variants in their native genomic context. For instance, endogenous levels of expression are maintained, and variants with disruptive effects on both splicing and protein function can be readily identified.

However, current methods for assaying pools of variants via genome editing all have limitations. Saturation genome editing

(SGE) leverages homology-directed DNA repair (HDR) to install hundreds of variants per experiment.[17] Yet suboptimal scalability arises from low HDR rates in many cell types as well as the requirement that variants be confined to a single region (100–200 bp) per experiment. Consequently, many separate SGE libraries are required to cover complete coding sequences.[16] Base editing screens are highly scalable and offer the advantage of being able to target sites genome wide,[18,19] yet are limited by the fact that most substitutions cannot be made by a single base editor and by the potential of unintended editing at target sites to confound results.

We reasoned that prime editing (PE) systems may constitute a way forward by enabling virtually any short variant in the genome to be installed.[20] Prime editors are Cas9 nickases coupled to reverse transcriptase domains that create programmed edits via reverse transcription (RT). A single prime editing guide RNA (pegRNA) determines both the site of nicking and the variant to be edited into the genome. Original PE systems displayed lower efficiencies than evolved base editors used in screening applications,[21] but recent innovations have boosted the performance of PE in human cells. These include engineered prime editors for greater activity,[22–24] improved design of pegRNAs,[25] and manipulation of host repair pathways to increase desired PE outcomes.[22]

Until recently, it remained unclear whether these improved PE systems would enable scalable functional characterization of variants genome wide. Saturation prime editing (SPE) was the first implementation of PE for assaying variant libraries.[26]

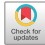

This approach was used to accurately identify pathogenic variants but required testing individual pegRNAs for activity prior to library design and direct sequencing of edited loci for effect quantification. More recent studies have demonstrated that PE screening using lentiviral delivery of pegRNAs offers the promise of introducing virtually any short variant in a cost-effective manner.[27–33] While this approach may prove ideal for testing large libraries of variants, work in this area has been restricted to positive selection screening or limited to specific edit types to mitigate the challenge of inefficient PE. Indeed, to what extent PE screens can accurately distinguish human variants of clinical relevance remains an open question.

Here, we optimize a platform for engineering large numbers of variants with PE in haploid human cells and rigorously assess the platform's potential to assay diverse variants in coding and non-coding regions using both negative and positive selection. We implement PE screening vectors that express two pegRNAs, thereby enabling a co-selection strategy to enrich for edited cells. Our findings show that with stringent filtering for highly active pegRNAs, variants leading to functional effects can be identified with reasonably high accuracy, suggesting that with further improvement and scaled implementation this platform will enable functional characterization of human variants genome wide.

## RESULTS

### A pooled PE screening platform for efficient installation of diverse edits

We sought to develop a high-throughput screening platform that enables the functional interrogation of diverse genetic variants installed via PE (Figure 1A). To this end, we created a monoclonal HAP1 line with stable expression of the optimized prime editor PEmax[22] via lentiviral integration (Figure 1B). HAP1 is a near-haploid human cell line with high value in genetic screening because recessive variant effects can be efficiently assayed.[34] Since HAP1 cells are mismatch repair (MMR) proficient, a second HAP1 line was generated with concomitant expression of a dominant negative MLH1 protein (MLH1dn) shown to enhance PE efficiency.[22]

We reasoned that akin to other pooled CRISPR screening modalities,[18,19,35] genomic integration and deep sequencing of pegRNAs over time could serve as a functional readout, provided each pegRNA efficiently introduces the variant it encodes. Such a system would afford highly scalable interrogation of variant effects without being confined to a single locus or gene per experiment.

Toward optimizing PE efficiencies, we leveraged engineered pegRNAs (epegRNAs).[25] Specifically, we incorporated the structured RNA motif tevopreQ$_1$ as a stabilizing 3′ extension cap. Hereafter, epegRNAs modified in this manner will be referred to simply as pegRNAs.

pegRNA design tools such as PRIDICT and DeepPrime[36,37] now offer predictions of editing efficiency for user-defined variants. However, our work predates these models, so we designed pegRNAs using PEGG.[28] To maximize our chances of including an efficient pegRNA for each variant, in our libraries we allowed up to nine different pegRNA designs per variant, with diverse spacer and 3′ extension sequences.

We reasoned that an integrated readout of pegRNA activity may be important for accurate variant scoring due to the large variability of PE efficiencies within pegRNA pools. Therefore, downstream of each pegRNA, we included 55 bp of genomic target sequence to serve as a surrogate target (ST), capable of informing pegRNA editing efficiency after genomic integration (Figure 1C). STs have been previously employed for pooled base editing[38] and (pe)gRNA activity screens[39,40] and were recently shown to have high value in the context of pooled PE screening of *TP53* variants.[28]

We experimentally tested the following strategies to further optimize variant installation via PE: (1) enrichment via co-selection; (2) use of optimized pegRNA scaffolds; and (3) incorporation of protospacer-adjacent motif (PAM)-disruptive synonymous edits in pegRNA design. Co-selection systems enable the enrichment of intended edits by co-editing a second locus leading to a selectable phenotype. This strategy has been shown to increase rates of intended edits in non-homologous DNA end joining (NHEJ)- and HDR-mediated editing,[41] as well as in base editing and PE.[42,43] In our system, we adopted a PE co-selection strategy in which an SNV within *ATP1A1*, a HAP1-essential gene, leads to resistance against the Na$^+$,K$^+$-ATPase inhibitor ouabain.[43]

First, we identified a pegRNA, which efficiently installs T804N with a silent, PAM-disruptive mutation in HEK293T cells (Figure S1A). We next determined editing and ouabain-selection performance with this pegRNA when expressed stably in our HAP1 PE line, observing 10.4% of cells with the intended T804N mutation at the earliest time point post transduction (Figure 1D). Continued culture of cells for 7 more days in the absence of ouabain resulted in a 3.3-fold increase in T804N editing to 34.6%, indicating that stable expression of all PE components leads to continuous accumulation of edits. After 7 days of ouabain selection, 84.4% of alleles contained both T804N and the silent PAM mutation. This experiment validates efficient *ATP1A1* editing and ouabain selection in our HAP1-based PE system.

Structurally stabilized guide RNA (gRNA) scaffolds have been shown to enhance various Cas-based activities.[44–46] We compared the performance of four different scaffold designs (Figures S1B–S1F) with a set of 973 pegRNAs installing 160 multinucleotide variants (MNVs) in *SMARCB1* (up to nine pegRNAs per variant). This experiment was conducted via lentiviral delivery of pooled pegRNA libraries to HAP1:PEmax+MLH1dn. Editing rates for each pegRNA-scaffold combination were assessed 7 days post transduction via deep sequencing of pegRNA-ST cassettes to quantify the percentage of ST reads containing correct edits (Table S1). Incorrect pegRNA-ST cassettes (e.g., due to recombination or errors in synthesis or sequencing) were discarded from analysis (STAR Methods and Figure S2).

pegRNAs with the original gRNA scaffold[47] achieved a median editing rate of 0.6% (Figure 1E). All stabilized scaffolds performed better than the original. The F+E scaffold, comprising an A-U flip and stem extension,[44] achieved a 2.4-fold improvement in median PE efficiency over the original scaffold (Figure 1F). 87% of all pegRNAs exhibited higher or equal editing

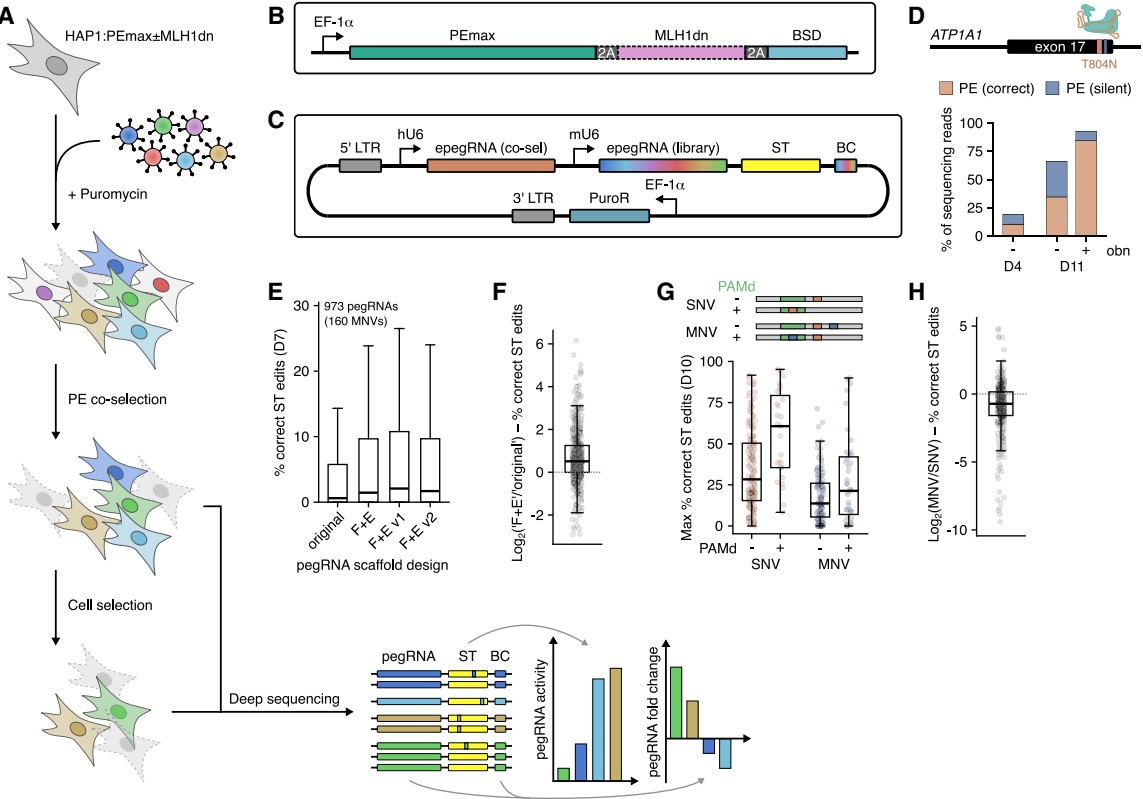

**Figure 1. Development of a pooled PE screening platform in HAP1**

(A) Schematic of the screening workflow, comprising lentiviral delivery of pegRNAs, puromycin selection, optional enrichment for edited cells via co-selection, functional selection (e.g., essentiality and drug treatment), deep sequencing of lentiviral cassettes, and quantification of pegRNA activities from surrogate targets (STs) and pegRNA frequency changes reflecting selective effects. Function scores for variants are determined from pegRNA frequency changes, using ST editing percentages to filter out inefficient pegRNAs.

(B) Schematic of the construct stably integrated into the HAP1 genome for 2A-coupled expression of PEmax, MLH1dn, and the blasticidin S resistance cassette, expressed from a minimal EF-1α promoter.

(C) Schematic of the vector for lentiviral delivery of pegRNA libraries. Each vector encodes a pegRNA to install a resistance mutation for co-selection and a pegRNA from the library to engineer a variant to be assayed. Each vector also includes a surrogate target (ST) matching the genomic sequence where the variant is to be introduced and a pegRNA-specific barcode (BC).

(D) Enrichment of cells with the PE-mediated T804N edit in exon 17 of *ATP1A1* in response to ouabain (obn) treatment, as determined by amplicon sequencing. Percentages of reads with correct PE (T804N) or partial PE (silent mutation only) are shown.

(E) Boxplot of ST editing percentages for 973 pegRNAs using different scaffolds.

(F) Boxplot of log$_2$ fold changes in ST editing percentages per variant using pegRNAs with the F+E scaffold compared to matched pegRNAs with the original scaffold (data from E).

(G) Schematic of SNV and MNV types showing target edit (orange) and additional silent mutation (blue). Boxplot of correct ST editing percentages for the top pegRNA per variant grouped by edit type and PAM disruption (PAMd).

(H) Boxplot of log$_2$ fold changes in ST editing percentages comparing MNV- and SNV-pegRNA pairs (data from G).

Boxplots: bold line, median; boxes, interquartile range (IQR); whiskers extend to points within 1.5× IQR; outliers not shown in (E). See also Figures S1 and S2; Tables S1 and S2.

activity using the F+E scaffold. Swapping the flipped A-U bases in the F+E scaffold with a G-C pair (v1)[45] led to a comparable increase in editing (median 2.1% correct ST editing; Figure 1E). Further stabilization by introducing a superstable loop within the first hairpin of the scaffold (v2), as reported in the t-lock design,[46] did not increase pegRNA activities. In summary, stabilized pegRNA scaffolds proved optimal for obtaining maximal PE efficiency across many pegRNA designs.

Next, we investigated whether SNVs are most efficiently introduced as individual mutations or with additional edits. Previous

work has suggested that programming silent mutations in addition to each target SNV may be advantageous because MNVs are less efficiently recognized by the MMR machinery.[22] To test this in our platform, we designed two pegRNA libraries, one encoding 191 distinct SNVs on their own and the other programming the same set of SNVs with one or two additional silent mutations, creating 166 corresponding MNVs in total. The libraries were pooled to a total of 2,188 pegRNAs and assayed in HAP1:PEmax+MLH1dn to measure ST editing rates for each pegRNA on day 10 (Table S2).

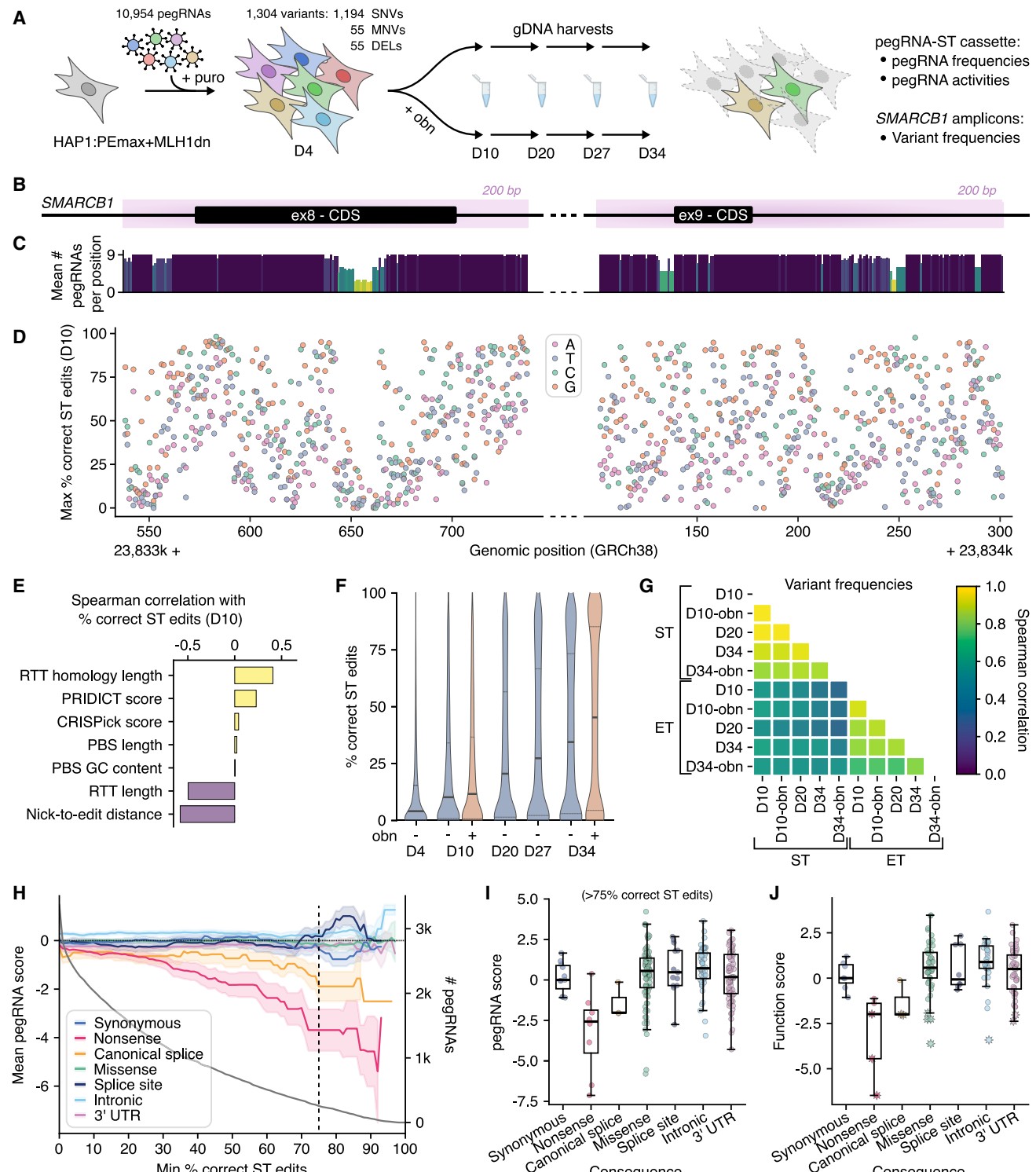

**Figure 2. Near-saturation mutagenesis of *SMARCB1* regions by pooled PE**

(A) Schematic of the experimental workflow for *SMARCB1* essentiality screening using pooled PE, comprising lentiviral delivery of pegRNAs to HAP1:PEmax+MLH1dn cells (D0), and culture until day 34 with sequential harvesting for deep sequencing of pegRNA-ST cassettes and *SMARCB1* target regions.

(B) Illustration of target regions (light purple) for saturation mutagenesis, centered on *SMARCB1* exons 8 and 9.

(C) For each genomic position, the average number of pegRNAs per SNV included in the library is plotted. pegRNA coverage per variant was capped at 9.

*(legend continued on next page)*

Only the best-performing pegRNAs per variant were analyzed to account for differences in how restrictive pegRNA design may be for either SNVs or MNVs. We confirmed the finding that edits within the PAM are more efficiently installed, with PAM-disruptive edits achieving 2.1- and 1.6-fold higher editing over other SNVs and MNVs, respectively (Figure 1G). Overall, however, we observed higher editing rates for SNVs than MNVs independent of PAM disruption (median 61% higher editing comparing SNVs to MNVs; Figure 1H). As most SNVs were installed more efficiently as single substitutions, we proceeded with screening libraries designed in this manner.

### High-throughput essentiality screening of *SMARCB1* variants

We next analyzed whether our pooled PE platform would enable the functional interrogation of variants via negative selection (Figure 2A). Specifically, we asked whether sequencing of pegRNA-ST cassettes alone could accurately define variants with deleterious effects (i.e., without requiring sequencing of endogenously installed variants).

*SMARCB1* encodes a core subunit of the BAF (BRG1/BRM-associated factor) chromatin remodeling complex, and inactivating mutations are known to drive diverse cancers.[48] *SMARCB1* is also among ~2,200 essential genes in HAP1,[49] meaning that LoF variants should be depleted in culture. We independently confirmed *SMARCB1* essentiality in HAP1 by observing depletion of frameshifting indels after Cas9-mediated editing (Figure S3A).

Exons 8 and 9 of *SMARCB1* fully encode the C-terminal coiled-coil domain (CTD), which harbors recurrently observed missense mutations in cancers and the rare developmental disorder Coffin-Siris syndrome.[48,50] We designed a pegRNA pool to install all possible SNVs in two 200-bp regions across exons 8 and 9 as well as MNVs encoding nonsense mutations and 3-bp deletions of every codon (Figure 2B).

We programmed variant-level redundancy in the pegRNA library, allowing a maximum of nine pegRNA designs per variant, which was possible for most SNVs (Figure 2C). Although pegRNA coverage for some mutations was lower due to the scarcity of NGG-PAM sequences, we successfully designed pegRNAs for nearly all intended SNVs (99.5%), MNVs (100%), and deletions (100%).

In total, 10,954 pegRNAs programming 1,194 SNVs, 55 MNVs, and 55 deletions were cloned into a lentiviral expression vector with the F+E scaffold. In addition to STs, the cassette included the co-selection pegRNA (*ATP1A1*-T804N) for optional enrichment of edited cells. The resulting library was introduced to HAP1:PEmax+MLH1dn via lentiviral transduction (multiplicity of infection [MOI] less than 0.5), and deep sequencing of pegRNA-ST cassettes was performed on days 4, 10, 20, 27, and 34 to measure pegRNA frequencies and editing dynamics. To test the co-selection strategy, one pool of cells was treated with ouabain on day 5, and two replicate pegRNA libraries without co-selection were also maintained (STAR Methods). We observed strong correlations for pegRNA frequencies and ST editing percentages between replicate libraries and across time points (Figures S3B and S3C).

We assessed PE efficiency of pegRNAs designed to introduce SNVs across both target regions (exons 8 and 9). First, to approximate variant-level editing activity from sequencing of pegRNA-ST cassettes, ST editing percentages were averaged across all pegRNAs encoding the same edit and mapped to the genomic coordinate of each variant. This revealed highly variable PE efficiencies within the pegRNA pool (Figure S3D). Forty percent of pegRNAs had ST editing rates above 10%. We also checked the top-performing pegRNA per SNV and found that 89.9% of all SNVs were installed with at least 10% ST editing by at least one pegRNA (Figure 2D). Editing rates across target regions were highly variable except for a few regions of lower PE efficiencies that coincided with low pegRNA coverage.

We next asked which pegRNA features correlated with higher PE efficiencies. Of pegRNA features analyzed, RT template (RTT) homology length (distance from edit to 5′ end of RTT) was most strongly correlated with ST editing on day 10 ($R = 0.43$, $r = 0.37$), followed by PBS length and PBS GC content (Figures 2E and S4A–S4G). Meanwhile, total RTT length and nick-to-edit distance were negatively correlated with ST editing. We also observed a modest correlation with CRISPick scores (i.e., predictions of gRNA on-target activity).[45,51] We retrospectively computed PRIDICT scores[36] where possible for our pegRNAs and found a reasonable correlation with experimentally determined activities, demonstrating the value of this tool for future pegRNA design. Notably, the features of pegRNAs most predictive of editing activity were highly consistent across libraries in this study (Figures S4H and S4I). Overall, these results are generally confirmatory of prior findings.[28,36,37] Minor differences, such

---

(D) The maximum ST editing rate for each SNV 10 days post transduction is plotted by position.

(E) Spearman correlation of correct ST editing percentages with different pegRNA features for a pool of 8,612 pegRNAs programming SNVs, MNVs, and deletions (DELs) across *SMARCB1*. ST editing percentages were averaged across day 10 duplicates without ouabain before correlation analysis.

(F) Violin plots of ST editing percentages observed across the pegRNA pool at multiple time points, with and without ouabain (obn) co-selection. Bold line indicates the median and dashed lines the interquartile ranges.

(G) Heatmap of Spearman correlation coefficients determined between surrogate target (ST) and endogenous target (ET) variant frequencies across samples. ST variant frequencies were computed as the fraction of all STs containing a given variant. ET variant frequencies were adjusted with a variant-level background correction based on negative control sample sequencing.

(H) Mean $\log_2$ fold change in pegRNA frequency (day 34 over day 10) in ouabain-treated samples by variant consequence. Mean values are plotted as a function of ST editing thresholds used for pegRNA filtering. Bands correspond to 95% confidence intervals, and the gray curve shows the number of pegRNAs passing frequency and ST editing thresholds. The dashed vertical line corresponds to 75% correct ST edits, chosen as the threshold for analysis in (I) and (J). pegRNAs with frequencies lower than $6 \times 10^{-5}$ on day 10 were excluded from analysis.

(I) pegRNA $\log_2$ fold change between day 34-ouabain and day 10-ouabain samples, grouped by consequence.

(J) pegRNA scores from (I), averaged per variant (function scores). Significantly scored variants ($q < 0.05$) are indicated with stars.

Boxplots: bold line, median; boxes, IQR; whiskers extend to points within 1.5× IQR. See also Figures S3–S6; Tables S3 and S4.

as the inverse correlation between RTT length and editing efficiency, likely stem from pegRNA design limitations imposed by capping synthetic oligo length at 243 nt.

One potential advantage of stably expressing all PE components is that intended edits may accumulate in culture over time. Indeed, for all libraries tested we observed a continued increase in ST editing with prolonged culture (Figure 2F). Median ST editing rates across the pegRNA pool increased from 4.1% on day 4 to 34% on day 34 for samples without co-selection. Overall, ST editing was highest with ouabain-mediated co-selection, reaching a median efficiency of 45% by day 34. Notably, ST editing percentages across all samples were bimodal by day 34. For instance, with co-selection, 33.0% of pegRNAs showed greater than 75% editing, while 30.6% of pegRNAs showed less than 10% editing.

For optimal scalability, ideally the presence of a variant in the genome could be inferred from sequencing of the pegRNA-ST cassette alone, but for this to work effectively, ST editing rates must accurately reflect endogenous target (ET) editing rates. To assess this, we sequenced ETs in *SMARCB1* amplified from the same cells used to measure pegRNA frequencies and ST editing. In the ouabain co-selected day-10 sample, we identified 79% of all intended variants at frequencies greater than background (defined via sequencing of unedited controls). Because multiple pegRNAs were used to engineer each variant, we used the variant-level ST editing rate calculated for each set of pegRNAs encoding a single variant for comparison to endogenous variant frequencies (Figures 2G and S5A), observing a correlation of $r = 0.53$ for all variants and $r = 0.69$ for MNVs and DELs. This demonstrates that ST editing can be used as a reasonable proxy for identifying pegRNAs that install variants with high activity.

Observing that nearly all variant frequencies in ETs slowly increased during the 34-day experiment, it was unclear whether selection against pegRNAs encoding LoF variants could be determined from sequencing pegRNAs (Figures S5B and S5C). To investigate this, we calculated a pegRNA score for each pegRNA in the library, defined as its day-34 frequency normalized to its day-10 frequency (Table S3). We also calculated variant-level function scores by averaging scores from pegRNAs installing the same variant (Table S4).

Given selection is predicted to be strongest against the most active pegRNAs encoding LoF variants, we explored how different ST editing activity filters impact our ability to separate signal from noise. If no threshold is applied, or only a lax threshold set to 10% ST editing on day 10, there is no clear separation between pegRNAs installing nonsense variants and synonymous variants (Figure 2H). However, using an ST editing threshold of 30%, pegRNAs encoding nonsense and canonical splice variants score distinctly lower than pegRNAs encoding synonymous variants. This trend becomes more pronounced if only highly active pegRNAs are retained, with a mean nonsense pegRNA score of −3.70 when using a stringent ST editing threshold of 75%. While effects of negative selection are readily apparent for highly active pegRNAs, imposing a stringent threshold excludes most pegRNAs tested, resulting in a considerable reduction in variants assayed (Figure 2H). Nevertheless, applying an ST editing threshold of 75% enables many variants predicted to be LoF (i.e., nonsense, canonical splice) to be distin-

guished by low pegRNA scores and low function scores (Figures 2I–2J and S5D–S5F).

Among 164 variants scored with this filter, we identified 12 significantly depleted variants, including three nonsense and two canonical splice variants, using a false discovery rate (FDR) of 0.05 to call LoF variants (STAR Methods). LoF missense variants clustered within a highly conserved α-helix of the SMARCB1 CTD. We also scored the intronic SNV c.1119−12C>G as LoF, which is annotated as a VUS in ClinVar[7] but has been hypothesized to disrupt splicing[52] and has a SpliceAI score[4] of 0.96.

To validate these findings, we selected eight significantly depleted variants for further analysis, including three missense variants, the intronic variant c.1119−12C>G, and two 3′-untranslated region (UTR) variants. These variants were reassessed in a newly generated PE7-expressing HAP1 line with *MLH1* knocked out (HAP1:PE7+MLH1-KO). Lentiviruses encoding pegRNAs programming each variant were mixed and assayed in a small pool (i.e., a "minipool") following the same procedure as for the large-scale screen. With this approach, we observed modestly higher ST editing rates for the same set of pegRNAs compared to before (day-4 median 78.2% versus 64.9% previously; Figure S6A). Function scores derived from pegRNA frequencies and endogenous variant frequencies with and without co-selection were highly correlated (Pearson's $r$ = 0.97–0.99; Figures S6B–S6D), and LoF effects of most but not all variants previously deemed LoF were confirmed, including one nonsense variant, one canonical splice variant, two of three missense variants, and the intronic VUS located in intron 8. Despite the reported pathogenicity of a nearby 3′-UTR variant in predisposition to schwannomatosis,[53] we did not observe LoF effects for the two 3′-UTR variants retested, indicating that they were likely false positives.

Additionally, we assayed the same variants individually by transducing HAP1:MLH1-KO cells with each pegRNA and transiently expressing PE7. This approach confirmed the LoF variants selected against in the minipool to be steadily depleted, in contrast to neutral variants (Figure S6E). Function scores determined by individually assaying variants correlated highly with function scores from the minipool experiment (Pearson's $r$ = 0.91; Figure S6F), further validating LoF effects. Lastly, given the clinical relevance of the intronic VUS, c.1119−12C>G, we assessed this variant's impact on splicing. Amplicon sequencing of *SMARCB1* transcripts revealed a novel splice junction unique to cells harboring c.1119−12C>G (Figure S6G). This new splice junction is immediately adjacent to c.1119−12C>G and leads to 11 bp of intron retention, corroborating this variant's effect at the molecular level.

Overall, these examples illustrate how select LoF variants efficiently installed by PE can be functionally identified via negative selection and suggest that further improvements to editing efficiency will enable many more variants to be assessed in this manner.

### Accurate determination of variant effects on *MLH1* function

We next used our platform to assay variants across *MLH1*, a tumor-suppressor gene encoding a subunit of the MMR pathway.

Germline pathogenic variants in *MLH1* predispose patients to several types of cancer, including colorectal and endometrial carcinoma, and nearly 2,000 VUSs in *MLH1* have been reported in ClinVar.[7] Unlike *SMARCB1*, *MLH1* is not essential in HAP1. However, it has been shown that loss of MMR pathway function in HAP1 leads to partial 6-thioguanine (6TG) resistance.[54] Therefore, 6TG selection may enable positive selection of pegRNAs encoding LoF variants. One advantage of this approach is that selection can be initiated after an extended period of editing, thus increasing the fraction of edited cells at the onset of selection.

To assess 6TG selection, we performed dilution series in wild-type (WT) HAP1, HAP1:PEmax+MLH1dn, and HAP1:PE2+ MLH1-KO cells (Figure S7A). Compared to WT, HAP1:PE2+ MLH1-KO cells were highly resistant to low doses of 6TG, whereas HAP1:PEmax+MLH1dn cells were only mildly resistant. Reasoning that use of HAP1:PEmax+MLH1dn cells for screening may lead to more efficient PE but less efficient selection, we performed screens in both HAP1:PEmax and HAP1:PEmax+ MLH1dn using different 6TG concentrations (1.2 μg/mL and 1.6 μg/mL, respectively). Screens were also performed both with and without ouabain co-selection.

Our first screen aimed to identify LoF SNVs in exon 10 of *MLH1*. This library consisted of 2,696 pegRNAs encoding 598 variants, including nearly all possible SNVs (96%) in a 200-bp region spanning exon 10 and flanking intronic regions, as well as 22 nonsense MNVs (Figure 3A). Libraries were designed, cloned, transduced, and sequenced as for *SMARCB1* experiments, with genomic DNA (gDNA) samples collected on days 4, 13, 20, and 34. Ouabain co-selection was performed from day 13 to day 20, and 6TG selection was initiated on day 20 prior to harvesting surviving cells on day 34. Sequencing of both pegRNA-ST cassettes and endogenous loci was performed.

We first compared ST editing and ET editing across cell lines and co-selection strategies. We observed higher editing at both STs and ETs in the HAP1:PEmax+MLH1dn cell line compared to HAP1:PEmax (Figure S7B). The median variant frequency was 1.5-fold higher in HAP1:PEmax+MLH1dn compared to HAP1:PEmax in cells without co-selection, and 1.9-fold higher in cells with co-selection. In both lines, ouabain co-selection modestly improved editing at day 20. For instance, the median ST editing rate on day 20 was 11.9% in ouabain-treated HAP1:PEmax+MLH1dn cells, compared to 9.1% in cells without ouabain. In the same line, the median variant frequency in ET sequencing increased from $2.4 \times 10^{-4}$ to $3.0 \times 10^{-4}$ with the addition of ouabain. As before, editing rates of individual pegRNAs were highly variable. For instance, 40% of pegRNAs had ST editing percentages over 25% in ouabain-treated HAP1:PEmax+MLH1dn cells.

To investigate whether LoF variants could be identified from changes in pegRNA frequencies over time, we calculated pegRNA scores as the $\log_2$ ratio of day-34 pegRNA frequency over day-20 pegRNA frequency (Table S5). As before, function scores for individual variants were obtained by averaging scores of all pegRNAs encoding the same variant with ST editing rates greater than a specified threshold (Table S6). To assess performance for identifying LoF variants, we defined synonymous variants as putatively neutral (pNeut) and nonsense and canonical splice variants as putatively LoF (pLoF), then calculated area under the receiver-operating characteristic curve (AUC) measurements using a continuous range of ST editing thresholds. This approach revealed thresholds that enable accurate detection of pLoF variants across all four screening conditions (Figure 3B). AUCs reached 1.00 in three of four screens using more stringent thresholds. For example, an AUC of 1.00 was reached using an ST editing threshold of 22% for pegRNAs assayed in HAP1:PEmax with co-selection (Figure 3B), while retaining 20% of pegRNAs designed.

To include more pegRNAs in analysis, relatively lax ST editing thresholds were set at 5% for experiments in HAP1:PEmax and at 25% for experiments in HAP1:PEmax+MLH1dn. While more stringent filters may perform optimally for a single condition, this approach allowed more pegRNA scores to be compared across screen conditions (Figure S8A).

We assessed how well pegRNA scores and function scores were correlated between conditions both before and after filtering pegRNAs on ST editing rates (Figures S9A–S9C). Correlations of pegRNA scores improved with filtering, in turn resulting in well-correlated function scores. To determine a final function score for each variant, we required the variant to have been assayed in at least two conditions and calculated the mean. This approach led to function scores for 401 out of 598 variants for which pegRNAs were designed, determined from 967 retained pegRNAs. To define LoF variants, we defined an empirical null distribution from synonymous variants and set the FDR to 0.01.

Mapping function scores to their genomic position reveals a cluster of LoF missense variants near the end of exon 10 (Figure 3C). These missense variants localize to a highly conserved β-sheet in the MLH1 structure (PDB: 4P7A; Figure 3D), revealing residues intolerant to mutation.

To orthogonally validate our exon 10 results, we compared function scores for missense variants to combined annotation-dependent depletion (CADD) scores, observing a Pearson's correlation of $r = 0.50$ (Figure 3E). Considering the possibility that MLH1dn expression may impact the effects of *MLH1* variants, we also computed cell-line-specific function scores using data from only HAP1:PEmax or only HAP1:PEmax+MLH1dn. However, correlations to variant effect predictors were strongest when data from both cell lines were combined (Figures S10A–S10C). Though rare, a small subset of LoF variants displayed stronger effects with MLH1dn expression (Figures S10D and S10E), for example, c.866A>G and c.871T>A.

As many variants in exon 10 have been reported in ClinVar, we also compared function scores across pathogenicity categories. While 57% of "pathogenic" and "likely pathogenic" variants in the region had function scores greater than 2.0, no "benign" or "likely benign" variants scored above 2.0 (Figure 3F). This suggests that our platform may enable scalable identification of new pathogenic variants. Indeed, 7.5% of VUSs tested in this region were LoF in our assay.

Depending on intended use, more stringent filtering of pegRNAs based on ST editing may be implemented. For example, we repeated analyses only on variants with average ST editing rates greater than 36%, corresponding to the lowest threshold at which AUC = 1.00 for distinguishing pLoF from

**Figure 3. Pooled prime editing of *MLH1* exon 10 identifies disease-associated variants**

(A) Experimental workflow of *MLH1* 6TG selection screens, comprising lentiviral delivery of pegRNAs to HAP1:PEmax and HAP1:PEmax+MLH1dn cells, ouabain co-selection (day 13–day 20), cell harvesting before (day 20) and after (day 34) 6TG challenge, and deep sequencing of pegRNA-ST cassettes and the *MLH1* target region.

(B) For each condition, AUC measurements for pLoF variant identification are plotted as a function of ST editing threshold. For this analysis, synonymous variants were defined as pNeut and nonsense and canonical splice variants as pLoF.

(C) Function scores for 401 variants are plotted by genomic position and colored by variant consequence. Significantly scored variants ($q < 0.01$) are indicated with stars.

(D) The highest function score of all missense variants assayed at each amino acid (AA) position is shown on the MLH1 structure (PDB: 4P7A).

(E) Correlation between function scores and CADD scores for 98 missense variants (Pearson's $r = 0.50$).

(F) Function scores for 79 variants present in ClinVar, grouped by pathogenicity annotation. Boxplot: bold line, median; boxes, IQR; whiskers extend to points within 1.5× IQR.

(G) Correlation between function scores (pegRNA-derived) and endogenous function scores for 205 variants (Pearson's $r = 0.69$).

See also Figures S7–S12 and S14; Tables S5 and S6.

pNeut variants. This approach improved the correlation to CADD scores ($r = 0.64$) and resulted in non-overlapping score ranges for five pathogenic SNVs and nine benign SNVs (Figure S11).

Finally, we performed sequencing of the endogenous *MLH1* exon 10 region to quantify variants' enrichment in genomic DNA following selection (Figures 3G and S12A–S12B). The highest-scoring variants, as determined by pegRNA-based function scores, were also highly enriched in endogenous

DNA. The correlation between function score and endogenous function score (defined as the $\log_2$ ratio of post-6TG frequency over pre-6TG frequency for each variant in ET sequencing) increased from 0.69 to 0.77 when analyzing only variants with average ST editing percentages greater than 36%. In summary, these experiments demonstrate successful identification of LoF variants with relatively high SNV coverage in a single genomic region while once more highlighting the importance of stringent pegRNA filtering.

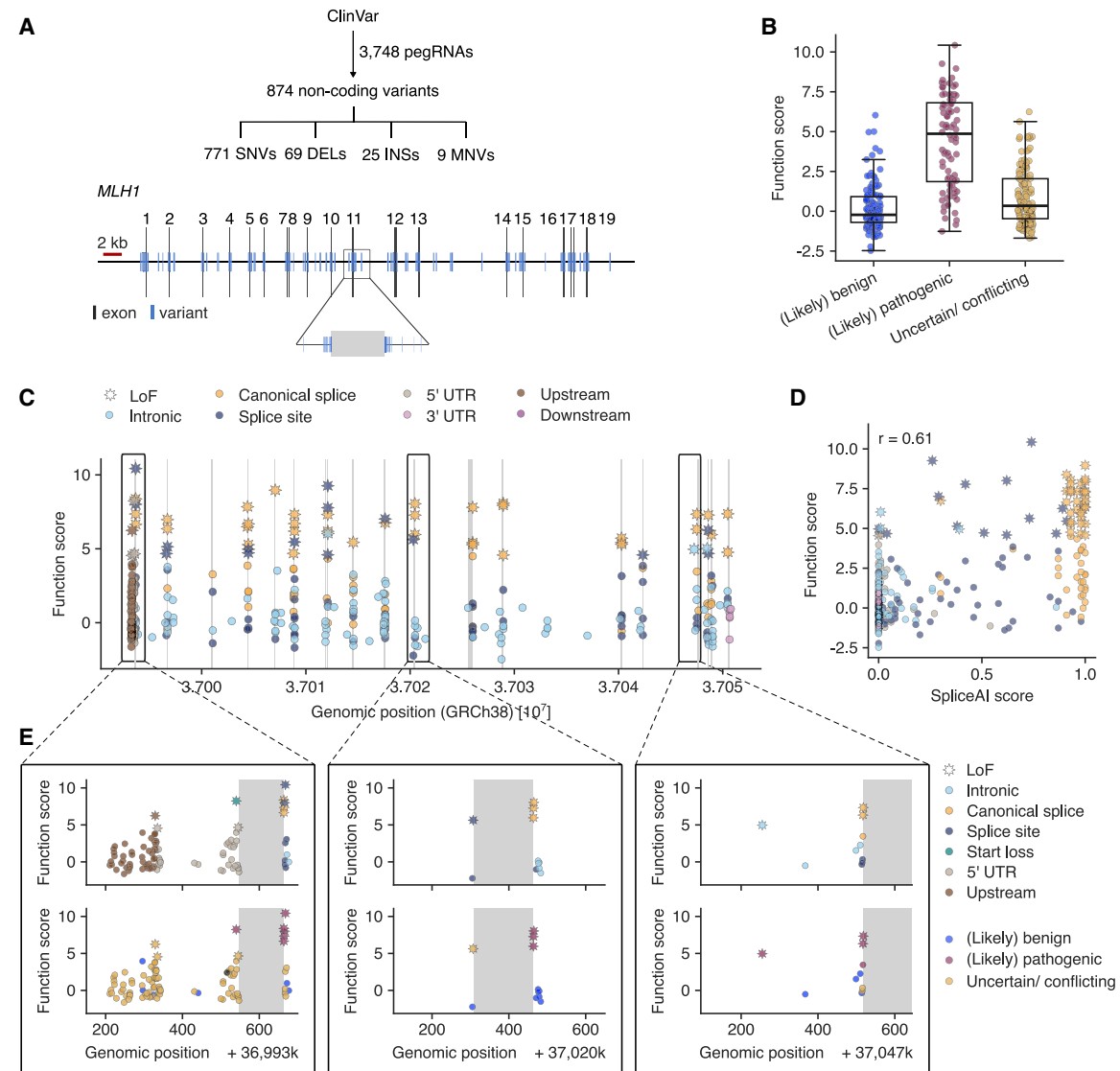

**Figure 4. Screening all non-coding *MLH1* variants in ClinVar for LoF effects**

(A) A library of 3,748 pegRNAs was designed to install 874 non-coding variants from ClinVar including 771 SNVs, 69 deletions (DELs), 25 insertions (INSs), and 9 MNVs. Variants, represented by blue lines, span the entire *MLH1* locus.

(B) Function scores are plotted for 357 variants, grouped by pathogenicity annotation in ClinVar. Boxplot: bold line, median; boxes, IQR; whiskers extend to points within 1.5× IQR.

(C) Function scores are plotted for 362 variants by genomic position and color-coded by variant consequence. Stars indicate variants for which $q < 0.01$.

(D) The correlation between function scores of SNVs and SpliceAI scores is plotted ($n = 296$).

(E) Detailed results for select regions are shown: exon 1 and upstream (left), exon 11 (middle), and intron 15 (right). Exonic regions are in gray, and variants are colored by consequence (upper panels) or ClinVar annotation (lower panels).

See also Figures S8–S10 and S12–S14; Tables S7 and S8.

## Screening all short non-coding *MLH1* variants in ClinVar in a single experiment

Non-coding variants are challenging to interpret clinically and have proven difficult to assay at scale via genome editing, especially when distributed over large sequence spaces. To evaluate whether our PE platform could be used for high-throughput assessment of non-coding variants, we designed a pegRNA library encoding all non-coding *MLH1* variants smaller than

10 bp and reported in ClinVar. This consisted of 3,748 pegRNAs covering 874 variants distributed across 60 kb of *MLH1*, with many variants located near intron-exon boundaries (Figure 4A).

Using the same strategy as for the *MLH1* exon 10 experiment, including repeating screens across four conditions and filtering pegRNAs by activity, we derived final function scores for 362 of the 874 variants tested (Tables S7 and S8). Compared to the results for exon 10, we observed modestly improved correlations

of pegRNA scores and function scores across conditions (Figures S9D–S9F). Once more, filtering of pegRNAs by ST editing was critical to see clear differences between pLoF variants and pNeut variants, defined for this experiment as canonical splice site variants and intronic variants more than 8 bp from any exon, respectively (Figure S8B). The screen that best discriminated pLoF variants was performed in HAP1:PEmax with ouabain treatment (Figure S13), again confirming an advantage of co-selection.

Importantly, the vast majority of ClinVar-pathogenic variants assayed had function scores greater than 1.5 (median = 4.87, SD = 2.90; Figure 4B). In contrast, nearly all benign variants scored neutrally (median = −0.22, SD = 1.45), while scores of VUSs spanned a wide range (median = 0.34, SD = 1.72). Using a stringent threshold to determine LoF variants ($q < 0.01$; Figure 4C), we called 54% of ClinVar-pathogenic variants as LoF and only 2.4% of ClinVar-benign variants as LoF (AUC = 0.89 for distinguishing pathogenic and benign).

As intronic variants deemed LoF are likely to impact splicing, we compared function scores to SpliceAI scores for orthogonal validation (Figure 4D). While the highest-scoring variants by SpliceAI tended to have high function scores, many variants with intermediate SpliceAI scores (0.25–0.75) also scored as LoF. Overall, only 2 of 96 intronic variants with SpliceAI scores less than 0.05 were LoF in our screen, suggesting predictive models of splicing will be valuable for prioritizing intronic variants for functional assessment. Whereas many LoF variants disrupt canonical splice sites, LoF variants were also observed in splice regions (3–8 bp from an exon), deeper in introns, in the 5′-UTR, and upstream of the transcriptional start site (TSS) (Figure 4E). Most variants in these regions lack definitive interpretations in ClinVar.

One such variant is c.885−3C>G, a VUS upstream of exon 11 predicted by SpliceAI to potentially cause acceptor site loss (SpliceAI score = 0.73). This variant scored comparably to known pathogenic variants adjacent to exon 11 in our screen (Figure 4E, middle). One notable variant deep within intron 15, c.1732−264A>T, is deemed "likely pathogenic" in ClinVar and scores intermediately by SpliceAI (0.39). This variant also scored as LoF in our assay (function score = 4.96; Figure 4E, right), corroborating its pathogenicity. We also scored c.−7_1del, an 8-bp deletion that disrupts the initiation codon as LoF, as well as 3.6% of other variants assayed in the 5′-UTR or upstream of the TSS.

To validate LoF effects of *MLH1* variants, we first performed amplicon sequencing of edited regions using the same gDNA from HAP1:PEmax cells from which pegRNAs were sequenced. Indeed, LoF variants including c.885−3C>G, c.1732−264A>T, and c.−7_1del were strongly enriched in their respective regions post 6TG selection, corroborating pegRNA-based function scores (Figures S12C and S12D). The lack of enrichment seen for nearly all SNVs in the 5′-UTR and upstream region was consistent with the low function scores of these variants. Notably, one variant in close proximity to the TSS, c.−219G>T, was deemed LoF in the screen but not strongly enriched in ET sequencing, indicating that its pegRNA-derived function score may be conflated by CRISPR inhibition or some other unintended effect.

We next selected 13 variants for further characterization using newly designed pegRNAs, including two neutral variants and two

variants strongly expected to be LoF. Other variants retested in this group included the start-loss variant (c.−7_1del), two splice region variants (c.884+3A>G and c.885−3C>G), the deep intronic variant c.1732−264A>T, the upstream variant c.−219G>T, two variants in the 5′-UTR (c.−213G>A and c.−3A>T), and two missense variants strongly enriched in HAP1:PEmax+MLH1dn but not in HAP1:PEmax (c.866A>G and c.871T>A). Lentivirus was produced for each pegRNA and combined to generate a minipool, which was then assayed in both HAP1 and K562 cells expressing PE7.

Across biological replicates and across both HAP1 and K562, ST editing rates and pegRNA selection were highly correlated (Figures S14A and S14B). Several variants that previously scored as LoF were again enriched upon 6TG treatment, including both expected LoF variants, the start-loss variant c.−7_1del, and both splice region variants (Figures S14C and S14D). Meanwhile, the upstream and 5′-UTR variants were not enriched in these experiments using independent pegRNAs, consistent with their lack of enrichment at the endogenous locus (Figure S12C) and confirming they were likely false positives in the larger screen. Notably, both missense variants retested were previously only enriched in HAP1 lines expressing MLH1dn (Figures S10D and S10E). In this experiment using HAP1 and K562 lines lacking MLH1dn, neither variant was enriched, suggesting that MLH1dn expression may impact how these variants score.

Among variants confirmed to be LoF, c.884+3A>G and c.885−3C>G both have intermediate SpliceAI scores of 0.29 and 0.73, respectively. Given the clinical importance of these variants, we further investigated their effects on splicing by individually transducing HAP1:PE7 cells with each corresponding pegRNA and enriching for successfully edited cells via 6TG selection. Gel electrophoresis and amplicon sequencing revealed that both variants drastically reduce canonical transcript isoforms and increase products of aberrant splicing (Figures S14E–S14G). Notably, c.884+3A>G has recently been linked to exon 10 skipping[55] and reclassified as likely pathogenic in ClinVar, consistent with our findings, whereas c.885−3C>G remains a VUS for which our functional data may prove valuable.

In summary, these experiments firmly establish that our PE platform can be used to identify LoF variants acting via diverse mechanisms across large non-coding regions. However, careful validation, for instance via ET sequencing or assaying of independent pegRNAs, is critical for improving the accuracy of screening.

## DISCUSSION

Here, we demonstrate a PE platform that enables scalable functional interrogation of coding and non-coding variants in haploid human cells. We optimize key components of this platform for more efficient editing, demonstrate negative selection against LoF variants in an essential gene, and score a total of 763 variants in *MLH1*, newly identifying LoF variants in both coding and non-coding regions. In the context of well-established MAVEs for assessing variants by genome editing, such as base editing screens,[18,19] SGE via HDR,[17] and SPE,[26] our platform offers the advantage of being able to install any set of short

variants virtually anywhere in the genome, provided an active pegRNA can be designed.

In our experiments, accurate variant scoring is made possible via inclusion of STs with each pegRNA assayed. The improvement in data quality observed upon filtering pegRNAs by ST editing reflects the fact that ST editing is a reasonable proxy for ET editing, as we have directly shown (Figures 2G and S5A). Notably, the same approach of using STs (i.e., "sensors") to filter inactive pegRNAs from analysis was also shown to improve PE screening results in a recent study of *TP53* variants.[28]

In our study, we prioritized testing modestly sized pegRNA libraries across different screening conditions and assays. However, we anticipate that assaying substantially larger pegRNA libraries will be readily achievable, as has been demonstrated for other PE platforms.[29] Based on our work, specific recommendations for performing future PE screens in HAP1 include: (1) allowing ample time for edits to accrue; (2) using a stabilized pegRNA scaffold, such as F+E v1; and (3) including STs to filter out inactive pegRNAs. Although we saw a consistent benefit of ouabain co-selection, increases in editing rates were modest, suggesting that this optimization may not always be necessary. Implementing recently described pegRNA scoring tools[36,37] in experimental design promises to improve coverage and reduce pegRNAs with low editing activities. Furthermore, newer prime editors, such as PE7,[24] may further improve editing rates. Whereas our data suggest that a small subset of *MLH1* variants may be influenced by expression of MLH1dn, for assaying variants in genes outside the MMR pathway, MLH1dn expression or *MLH1* KO will likely increase editing without confounding interpretation.

While the strong selection for LoF variants afforded by 6TG proved valuable for scoring *MLH1* variants accurately, ultimately additional functional assays will enable variants to be studied in more genomic regions. By establishing requirements for identifying LoF variants via negative selection, we illustrate a path forward for screening variants in over 2,000 genes that are essential in HAP1.[49] We envision that this may be particularly valuable for prioritizing sets of variants identified in clinical sequencing for further study and for testing effects of variants predicted to act via specific mechanisms across a large number of genes.

Our work illustrates the importance of validating functional effects of pegRNAs to confirm variant effects, as both false positives and false negatives can occur in pegRNA screening. We used multiple approaches to validate variant effects, including sequencing ETs across time points, replicating results with orthologous pegRNAs tested in small batches or individually, and confirming effects across multiple cell lines. Going forward, improved pegRNA design may allow multiple, orthogonally designed pegRNAs per variant to be tested, either together or in successive experiments. For assaying variants observed in patients, there will remain a need for rigorous clinical benchmarking prior to using data to aid variant classification.

We predict that implementing the key improvements we have outlined will ultimately make data from genome-wide PE screens valuable for variant classification, as has been established for other MAVEs. Further, scaling this framework to test computationally prioritized sets of variants may yield data suited for training and refining predictive models. By presenting a means of engineering and assaying select human variants across large sequence spaces, our HAP1-based PE platform promises to substantially improve identification of variants underlying human disease.

## Limitations of the study

Current limitations of our platform stem largely from only a limited fraction of pegRNAs leading to robust editing. This explains the incomplete scoring of variants for which we designed pegRNAs. As we show for variants in both *SMARCB1* and *MLH1*, separation of signal from noise depends strongly on discerning active pegRNAs via ST editing, as editing percentages among pegRNAs vary highly. In these proof-of-concept experiments, we also observed higher false-negative rates for expected LoF variants than demonstrated in many recent MAVE implementations.[16,54] We show that this effect can be mitigated via more stringent pegRNA filtering, although this comes at a cost of reduced coverage.

Importantly, as PE reagents continue to improve and pegRNA design tools mature, we anticipate a larger fraction of pegRNAs in future experiments will produce high-quality data. Many variants throughout the *MLH1* gene had mildly positive function scores but did not pass the FDR cutoff ($q < 0.01$) and therefore were not deemed LoF. It remains to be determined whether such variants may be hypomorphic alleles or whether relatively low endogenous editing rates preclude stronger selection for certain pegRNAs. In the future, the precision of function scores may be improved by incorporating strength of pegRNA selection and ST editing into variant scoring. Additionally, measuring effects of multiple, highly active pegRNAs per variant promises to improve score resolution.

### RESOURCE AVAILABILITY

#### Lead contact
Further information and requests for resources and reagents should be directed to and will be fulfilled by the lead contact, Gregory M. Findlay (greg.findlay@crick.ac.uk).

#### Materials availability
- Plasmids generated in this paper intended for common use will be made available on AddGene: pLenti_hU6-pegRNA(ATP1A1-T804N)_mU6-BsmBI_Puro: #234070, pLenti_PEmax-2A-BSD: #234071, pLenti_PEmax-2A-MLH1dn-2A-BSD: #234072, pLenti_PE7-2A-BSD: #234073.
- Additional constructs are available from the lead contact upon request.

#### Data and code availability
All experimental data including pegRNA frequencies, pegRNA scores, ST editing rates, and function scores are available in Supplemental Information. Raw NGS data (.fastq files) have been deposited (European Nucleotide Archive: PRJEB85691). Custom scripts available on GitHub (https://github.com/FrancisCrickInstitute/PooledPEScreen) have been minted with Zenodo (https://doi.org/10.5281/zenodo.14843845).

### ACKNOWLEDGMENTS

We thank the Crick's Genomics STP for performing sequencing, the Crick's Cell Services STP for maintenance of cell lines, and Joachim De Jonghe for providing reagents. This work was supported by the Francis Crick Institute, which receives its core funding from Cancer Research UK (CC2190), the UK Medical Research Council (CC2190), and The Wellcome Trust (CC2190). For the purpose of Open Access, the authors have applied a CC BY public copyright license to any Author Accepted Manuscript version arising from this submission.

## AUTHOR CONTRIBUTIONS

M.H., C.M.K., and G.M.F. conceived the project and designed experiments. C.M.K. and M.H. performed experiments and analyzed the data. M.H., C.M.K., M.B., A.C., M.S., and G.M.F. performed initial optimizations and derived key reagents. G.M.F. supervised the project. M.H., C.M.K., and G.M.F. wrote the manuscript with input from all authors.

## DECLARATION OF INTERESTS

The authors declare no competing interests.

## STAR★METHODS

Detailed methods are provided in the online version of this paper and include the following:

- KEY RESOURCES TABLE
- EXPERIMENTAL MODEL DETAILS
  - Cell lines and culture
- METHOD DETAILS
  - Plasmids
  - *SMARCB1* indel depletion in HAP1
  - pegRNA library design
  - pegRNA library cloning
  - Lentivirus production and titering
  - Generation of PE cell lines
  - *ATP1A1*-T804N pegRNA optimisation
  - *ATP1A1*-T804N enrichment with ouabain
  - 6TG dose titration
  - Pooled PE screening
  - pegRNA scaffold activity screen
  - SNV versus MNV pegRNA activity screen
  - *SMARCB1* saturation mutagenesis screen
  - *MLH1* exon 10 and non-coding variant screens
  - Amplicon sequencing of gDNA
  - Variant effect validation experiments
  - *SMARCB1* variant effect validation
  - *MLH1* variant effect validation in HAP1:PE7 cells
  - *MLH1* variant effect validation in K562:PE7 cells
  - Detection of aberrant splice products
  - Extraction of genomic DNA and PCR amplification
  - Illumina sequencing
- QUANTIFICATION AND STATISTICAL ANALYSIS
  - Sequencing data analysis
  - ET analysis
  - pegRNA-ST read processing
  - Calculation of pegRNA scores and function scores
  - AUC analysis
  - Comparing function scores with orthologous datasets
  - Analysis of sequencing data from validation experiments

## SUPPLEMENTAL INFORMATION

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

## Article

**CellPress**

## STAR★METHODS

### KEY RESOURCES TABLE

| REAGENT or RESOURCE | SOURCE | IDENTIFIER |
| --- | --- | --- |
| **Bacterial and virus strains** | | |
| NEB Stable Competent E. Coli | NEB | C3040H |
| Electrocompetent Endura Cells | Biosearch Technologies | 60242–2 |
| **Chemicals, peptides, and recombinant proteins** | | |
| 10-deacetylbaccatin III | Stratech | A4395-APE |
| Puromycin (dihydrochloride) | Cayman Chemical | 13884 |
| Blasticidin S (hydrochloride) | Gibco | R21001 |
| Ouabain (octahydrate) | Sigma-Aldrich | O3125 |
| 6-Thioguanine | Sigma-Aldrich | A4882 |
| **Critical commercial assays** | | |
| Xfect Transfection Reagent | Takara Bio | 631318 |
| DNeasy Blood and Tissue Kit | Qiagen | 69506 |
| KAPA HiFi HotStart ReadyMix | Roche | KK2602 |
| NEBuilder HiFi DNA Assembly Master Mix | NEB | E2621L |
| **Deposited data** | | |
| Screening data fastq files | This paper | ENA: PRJEB85691 |
| **Experimental models: Cell lines** | | |
| HEK293T | ATCC | CRL-3216 |
| K562 | ATCC | CCL-243 |
| HAP1 | Horizon Discovery | C631 |
| HAP1:PEmax-2A-BSD | This paper | N/A |
| HAP1:PEmax-2A-MLH1dn-2A-BSD | This paper | N/A |
| **Oligonucleotides** | | |
| Custom oligos and gene fragments | This paper | See Table S9 |
| **Recombinant DNA** | | |
| pLenti_hU6-peg(*ATP1A1*-T804N)_mU6-BsmBI_Puro | This paper | AddGene: 234070 |
| pLenti_PEmax-2A-BSD | This paper | AddGene: 234071 |
| pLenti_PEmax-2A-MLH1dn-2A-BSD | This paper | AddGene: 234072 |
| pLenti_PE7-2A-BSD | This paper | AddGene: 234073 |
| **Software and algorithms** | | |
| PEGG | Gould et al. 2024[28] | https://github.com/samgould2/PEGG2.0 |
| CRISPResso2 | Clement et al. 2019[56] | https://github.com/pinellolab/CRISPResso2 |
| DiMSum | Faure et al. 2020[57] | https://github.com/lehner-lab/DiMSum |
| Custom analysis scripts | This paper | https://github.com/FrancisCrickInstitute/PooledPEScreen |

## EXPERIMENTAL MODEL DETAILS

### Cell lines and culture

HEK293T (female, CRL-3216) and K562 (female, CCL-243) cells were obtained from American Type Culture Collection (ATCC). HAP1 cells (male lacking Y chromosome, C631) were obtained from Horizon Discovery.

HEK293T cells were cultured in Dulbecco's Modified Eagle Medium (DMEM) (#10564011, *Gibco*) supplemented with 10% fetal bovine serum (FBS) (#A5256701, *Gibco*) and 1% Penicillin-Streptomycin (Pen-Strep) (#15140122, *Gibco*). Cells were cultured at 37°C with 5% $CO_2$ and passaged using Versene solution (#15040066, *Thermo Fisher*). HAP1 cells were cultured in Iscove's Modified Dulbecco's Medium (IMDM) (#12440053, *Gibco*) supplemented with 10% FBS, 1% Pen-Strep, and 2.5 μM 10-deacetylbaccatin III (DAB)

(#A4395-APE, *Stratech*) at 37°C with 5% $CO_2$. Cells were passaged every 2–3 days using 0.25% trypsin-EDTA (#25200056, *Gibco*) to keep them below 80% confluency. K562 cells were cultured at 37°C with 5% $CO_2$ in Roswell Park Memorial Institute (RPMI) medium (#11875093, *Gibco*) supplemented with 10% FBS and 1% Pen-Strep. Cell density was maintained below $0.8 \times 10^6$ cells/mL. For culture of PE-expressing HAP1 and K562 lines, media was additionally supplemented with 5 μg/mL or 10 μg/mL blasticidin S (R21001, *Gibco*), respectively. All cell lines were confirmed to be free of mycoplasma.

## METHOD DETAILS

### Plasmids
For stable integration of PEmax with and without MLH1dn, a lentiviral vector was prepared by PCR amplification of the corresponding cassettes from pCMV-PEmax-P2A-hMLH1dn (#174828, *AddGene*) and subsequent insertion into pLenti-PE2-BSD (#161514, *AddGene*), which had been digested with EcoRI and XbaI (#R0145S & #R3101S, *NEB*). A lentiviral construct for stable integration of PE7 (pLenti-PE7-2A-BSD) was created by digestion of pLenti_PEmax-2A-BSD with BamHI and XbaI (#R0145S & #R3136S, *NEB*). PEmax was replaced with PE7, which was amplified from pT7-PE7 for IVT (#214813, *AddGene*), via Gibson assembly. For transient expression of PE7, pCMV_PE7-2A-BSD was constructed by Gibson assembly of a PCR-linearized pCMV-PEmax-P2A-hMLH1dn backbone with the PCR-amplified PE7-2A-BSD cassette from pLenti-PE7-2A-BSD. Single pegRNAs were stably integrated using lentiviral vectors cloned by assembling pegRNA oligo duplexes into BsmBI-digested (#R0739L, *NEB*) Lenti_gRNA-Puro (#84752, *AddGene*). The dual-pegRNA lentiviral vector, used for pegRNA library cloning, was prepared by amplification of a gene fragment encoding the mU6 promoter and a BsmBI cloning site (*Twist Bioscience*) followed by assembly into a KflI- and Eco72I- digested (#FD2164 & #FD0364, *Thermo Scientific*) Lenti_gRNA-Puro plasmid with a pre-inserted co-selection pegRNA under the hU6 promoter. For bacterial amplification of pegRNA scaffolds, oligo duplexes, encoding the different scaffold designs (original, F + E, F + E v1, F + E v2) and flanking BsmBI recognition sites, were inserted into the KpnI-digested (#R3142S, *NEB*) pU6_pegRNA-GG-Acceptor plasmid (#132777, *AddGene*) via Gibson assembly. All constructs were propagated in NEB Stable Competent *E. coli* (High Efficiency; #C3040H, *NEB*) unless stated otherwise.

### *SMARCB1* indel depletion in HAP1
Approximately $4 \times 10^6$ HAP1 cells were transfected using Xfect Transfection Reagent (#631318, *Takara Bio*) for each pSpCas9(BB)-2A-Puro construct (#62988, *AddGene*). 24h after transfection, the culture media was exchanged and supplemented with 1 μg/mL puromycin (#13884, *Cayman Chemical*). Approximately $10^7$ cells were harvested on days 6, 11, 15, 20, and 26 after transfection and stored at −80°C. Cell pellets were processed for extraction and purification of genomic DNA using DNeasy Blood and Tissue Kit (#69506, *Qiagen*). Target loci were amplified from purified gDNA by PCR using NEBNext Ultra II Q5 Master Mix (#M0544X, *NEB*). Adapters for dual-indexed Illumina sequencing were attached in a second PCR step. Sequencing reads were processed using the Cas9 mode of CRISPResso2 and its default parameters.[56]

### pegRNA library design
All pegRNA libraries were designed using a Jupyter Notebook implementation of the open-source python package PEGG.[28] For *SMARCB1* and *MLH1* saturation mutagenesis libraries, all possible SNVs in a 200 bp window centered on each target exon were created. We additionally introduced nonsense mutations at each codon via multinucleotide substitutions. For *SMARCB1* saturation mutagenesis libraries, we also included single-codon deletions. MNVs for the experiment comparing editing rates of SNVs and MNVs were designed from a random subset of the SNVs programmed as part of the *SMARCB1* saturation mutagenesis experiment. Up to two additional synonymous mutations were programmed in the four neighboring codons nearest the target codon. ClinVar variants within *MLH1* were accessed on 11/29/2022, filtering for short variants (less than 10 bp) with variant start positions mapping to non-coding regions.

 *SMARCB1*-targeting libraries were designed to include up to 9 pegRNAs per variant, while *MLH1* libraries contained up to 12 pegRNAs per variant. To diversify the set of pegRNAs per variant, we enforced the use of distinct spacer sequences (3 distinct spacers for *SMARCB1* pegRNAs and 2 distinct spacers for *MLH1* pegRNAs, respectively) and selected the 3 top-scoring pegRNAs within each set of pegRNA designs using a common spacer. All pegRNAs were appended with the tevopreQ$_1$ motif followed by a T$_7$ termination sequence. Additionally, each pegRNA was coupled to a 55-nt surrogate target (ST) sequence (replicating the endogenous target) and a unique 16-nt pegRNA barcode. The scaffold sequence was replaced with a BsmBI cloning cassette for ordering oligos, and pegRNA oligos were divided into multiple sub-libraries, each with a unique 10-nt library barcode at the oligo's 3′ end to allow specific amplification of individual libraries from the oligo pool. To create oligos of equal size (243 nt), a stuffer sequence was inserted between the T$_7$ termination sequence and ST sequence. Finally, designed pegRNA oligos containing BsmBI recognition sites that would interfere with cloning were discarded. pegRNA oligo libraries were ordered and synthesized as a custom oligo pool (*Twist Bioscience*).

### pegRNA library cloning
pegRNA libraries were cloned from oligo pools into lentiviral vectors in a two-step procedure. pegRNAs were amplified from the oligo pool via PCR using KAPA HiFi HotStart ReadyMix (#KK2602, *Roche*). pegRNA oligo subsets were specifically amplified from the oligo pool using primers specific for the library BC (Table S9). Thermocycling was performed according to guidelines for KAPA HiFi

HotStart ReadyMix except for elongated extension steps of 2 min. PCR products were purified and concentrated using AMPure XP SPRI Reagent (#A63881, *Beckman Coulter*) and subsequently by agarose gel electrophoresis to isolate 260 bp amplicons.

The dual pegRNA lentiviral vectors with pre-integrated co-selection pegRNAs were prepared for pegRNA library cloning by BsmBI-digestion and Quick CIP (#M0525L, *NEB*) treatment followed by gel purification. pegRNA oligo amplicons were assembled with the cut lentiviral vector via NEBuilder HiFi DNA Assembly Master Mix. The assembled plasmid pool was purified and concentrated with AMPure XP SPRI Reagent and used for transformation of Electrocompetent Endura Cells (#60242-2, *Biosearch Technologies*) following manufacturer's instructions and a previously published protocol.[58] The plasmid pool was extracted and purified using ZymoPURE II Plasmid Maxiprep Kit (#D4202, *Zymo Research*).

A second cloning step was next required to insert a pegRNA scaffold sequence. Therefore, the plasmid pool was subjected to BsmBI-digestion, Quick CIP treatment, and gel-electrophoretic purification. Separately, pegRNA scaffolds were prepared for cloning by BsmBI-digestion of pScaffold plasmids followed by gel purification. The digested vector library and the pegRNA scaffold were ligated using T4 DNA Ligase (#M0202, *NEB*). SPRI-purified and concentrated ligation product was used for transformation of Electrocompetent Endura Cells, as before, and plasmid pools were purified for lentivirus production. A 1,000-fold coverage of pegRNA library size was ensured at each transformation step, with the exception of the *MLH1* non-coding library, where the minimum coverage exceeded 250-fold at each step.

For cloning of the pegRNA pool to compare performance of different scaffolds, the first step of pegRNA library cloning remained the same. Scaffolds for all tested designs were prepared from corresponding pScaffold plasmids via BsmBI-digest and pooled at equimolar ratios. This scaffold mix was used in the subsequent ligation reaction with the BsmBI-digested plasmid pool before proceeding as detailed above.

The pegRNA pool for the *SMARCB1* variant screen was cloned into two lentiviral vectors with distinct co-selection pegRNAs. One vector encoded the *ATP1A1*-T804N pegRNA for ouabain co-selection, while the other vector contained a *HPRT1*-A161E pegRNA. Lentiviral particles were produced from both plasmid pools, which were subsequently used for transductions in *SMARCB1* variant screening experiments. Although intended to provide an alternative co-selection strategy, the pegRNA pool in the *HPRT1*-A161E co-selection vector served only as a duplicate library for assaying variants without co-selection.

### Lentivirus production and titering

For lentivirus production, each transfer plasmid was mixed with packaging and envelope plasmids (pLP1, pLP2, VSV-G) to 33 μg total DNA which was used for transfection of $2 \times 10^7$ HEK293T cells in a 15-cm culture dish with Lipofectamine 2000 (#11668019, *Invitrogen*). Viral supernatants were collected 2 days and 3 days post-transfection, before pooling and concentrating with PEG-8,000 (#V3011, *Promega*). Aliquots of concentrated lentivirus were stored at −80°C until used. Aliquots of lentivirus particles were titered by ddPCR as described.[59]

### Generation of PE cell lines

HAP1:PEmax and HAP1:PEmax+MLH1dn cell lines were generated via transduction of $3 \times 10^6$ HAP1 cells at low MOI (less than 0.3) with lentiviral particles packaged using pLenti_PEmax-2A-BSD transfer plasmids with and without MLH1dn expression. 2 days after transduction, the media was replaced with media containing 5 μg/mL blasticidin. Selection of transduced cells was performed for 14 days before expanding single clones. Final clones were chosen based on functional validation of PE activity via *ATP1A1* editing followed by ouabain selection. Complete genomic integration of the PEmax+MLH1dn sequence was validated via PCR of this cassette from genomic DNA followed by gel electrophoresis.

The HAP1:PE2+MLH1KO line was used only for titering 6TG dose. It was created by first transducing parental HAP1 cells with pLenti-PE2-BSD at low MOI (as for PEmax cell lines), then transfecting successfully transduced cells with a pX459 construct targeting *MLH1*. Individual clones were isolated and screened by Sanger sequencing, and a line with a frameshifting indel in exon 18 was selected. The HAP1:PE7+MLH1KO cell line used for validation experiments was created by first transfecting parental HAP1 cells with a pX459 construct targeting *MLH1*, followed by sequencing to confirm a clonal population with a 1 bp insertion in *MLH1* exon 10. This clone, HAP1:MLH1KO, as well as parental HAP1 and K562 cells were then transduced with pLenti_PE7-2A-BSD. Spinfection of K562 cells was performed by centrifugation of virus with cells at $1,321 \times g$ for 1.5 h at 37°C. Cells were selected for 10 to 14 days with 5 μg/mL (HAP1:PE7) or 10 μg/mL (K562:PE7) blasticidin S and subsequently used for variant effect validation experiments.

### *ATP1A1*-T804N pegRNA optimisation

Around $10^5$ HEK293T cells were co-transfected with pCMV-PEmax-P2A-hMLH1dn and pU6_pegRNA plasmids (6 different pegRNA designs) using Xfect Transfection Reagent. 4 days after transfection, cells were harvested and gDNA was extracted and purified. The target locus was amplified from gDNA by PCR using NEBNext Ultra II Q5 Master Mix (#M0544L, *NEB*). Adapters for dual-indexed Illumina sequencing were attached in a second PCR step. Sequencing reads were processed using the prime editing mode of CRISPResso2 to determine the fraction of reads with desired edits.

### *ATP1A1*-T804N enrichment with ouabain

Approximately $10^7$ HAP1:PEmax+MLH1dn cells were transduced at low MOI (approximately 0.1) with lentivirus particles produced from a Lenti_pegRNA-Puro plasmid encoding the *ATP1A1*-T804N pegRNA under the hU6 promoter. Cell selection was initiated 1 day

later by supplementation of the culture media with 1 μg/mL puromycin and was maintained for 3 days. On day 4 post-transduction, the cell pool was split in two for continued culture with and without ouabain treatment. Selection of cells with the *ATP1A1*-T804N edit was started by addition of 5 μM ouabain (#O3125, *Sigma-Aldrich*) and maintained until the end of the experiment. Cells were harvested for gDNA extraction and purification on days 4 and 11 post-transduction. The target locus was PCR amplified and sequenced by NGS. The fraction of sequencing reads with perfect and partial editing were determined via counting matches for corresponding 18-nt subsequences and dividing by total number of sequencing reads.

### 6TG dose titration

$1.2 \times 10^7$ HAP1:PEmax, HAP1:PEmax+MLH1dn, and HAP1:PE2+MLH1KO cells were treated for 6 days with a range of 6TG (#A4882, *Sigma-Aldrich*) concentrations: 0μM, 0.5μM, 0.8μM, 1μM, 1.2μM, 1.5μM, 1.8μM, 2μM, 2.5μM, 3μM, 3.5μM, and 4μM. Viable cells post drug challenge were counted using the Vi-Cell analyzer (*Beckman*).

### Pooled PE screening

For all pooled PE screens, prepared pegRNA lentivirus aliquots were thawed from frozen and used to transduce HAP1:PEmax lines at low MOIs (0.1–0.5), maintaining an average pegRNA coverage of at least 1,000-fold for each library. As a negative control, we also transduced HAP1 cells not expressing PEmax at a higher MOI of approximately 10, achieving 100× pegRNA coverage. One day after transduction, the culture media was exchanged and supplemented with 1 μg/mL puromycin for 3 days of selection, at which point MOIs were confirmed by examining cell confluency. Aliquots of cells were harvested periodically throughout the experiment, ensuring at least 1,000-fold average coverage of the library at each timepoint. Negative control samples were harvested at the earliest timepoint (day 4 or day 5) and served to determine background variant rates in STs. Additional experimental details for each pooled PE experiment are as follows.

### pegRNA scaffold activity screen

Approximately $4 \times 10^7$ HAP1:PEmax+MLH1dn cells were treated with lentivirus particles, achieving an MOI of approximately 0.1. The negative control sample (transduced HAP1 cells without PEmax expression) was harvested on day 5, while the experimental pool was harvested on day 7 to allow additional time for editing.

### SNV versus MNV pegRNA activity screen

pegRNA library cloning, lentivirus production, and screening were carried out in duplicate. The pegRNA library was tested in HAP1:PEmax+MLH1dn cells as part of the larger *SMARCB1* variant screen and analyzed separately. Comparison of SNV and MNV ST editing rates was performed using non-ouabain-treated samples harvested on day 10, whereas the negative control sample was processed on day 4.

### *SMARCB1* saturation mutagenesis screen

*SMARCB1* pegRNA library cloning, lentivirus production, and transduction of HAP1:PEmax+MLH1dn cells were performed in duplicate (once cloned into a vector co-expressing the *HPRT1*-A161E pegRNA and once cloned into a vector co-expressing the *ATP1A1*-T804N pegRNA for ouabain co-selection). To achieve at least 1,000-fold average coverage of the pegRNA library ($n = 12,211$ pegRNAs), total cell numbers were maintained above $1.3 \times 10^7$ throughout the experiment. After completion of puromycin selection 4 days post-transduction, the cell pools were split into two. One pool was maintained in media containing 5 μM ouabain while the remaining two duplicates were left untreated. Cells were passaged every 2–3 days and pellets of at least $3 \times 10^7$ cells were harvested on days 4, 10, 20, 27, and 34 post-transduction. The negative control sample was processed on day 4.

### *MLH1* exon 10 and non-coding variant screens

The same experimental procedure was applied for the *MLH1* exon 10 saturation mutagenesis and non-coding ClinVar variant screens. Lentiviral aliquots of pegRNA pools were used for transduction of both HAP1:PEmax and HAP1:PEmax+MLH1dn cell lines (D0). To achieve at least 1,000-fold average coverage of the pegRNA library ($n = 2,696$ pegRNAs for the exon 10 pool and $n = 3,748$ pegRNAs for the non-coding ClinVar pool), at least $4 \times 10^6$ cells were maintained throughout the experiment and harvested at each sampling. Puromycin selection was completed by day 4 and on day 13 the cell pools were split into two. One cell pool per cell line was treated with 5 μM ouabain while the other was left untreated, resulting in a total of four cell pools per experiment (HAP1:PEmax and HAP1:PEmax+MLH1dn each with and without ouabain co-selection). 20 days post-transduction, 6TG was added to the culture media at a concentration of 1.2 μg/mL for HAP1:PEmax and 1.6 μg/mL for HAP1:PEmax+MLH1dn. Cells were passaged every 2–3 days with addition of fresh selection media until day 34. Cell pellets of at least $10^7$ cells were harvested on day 4, day 20 (pre-selection), and day 34 (post-selection). The negative control sample was harvested on day 4.

### Amplicon sequencing of gDNA

Cell pellets were processed for gDNA extraction and deep sequencing of both pegRNA-ST cassettes and ETs was performed. For *SMARCB1* and *MLH1* exon 10 saturation mutagenesis screens, both pegRNA-ST cassettes and ETs were sequenced, while for screens in which only pegRNA activity was assessed only the pegRNA-ST cassette was sequenced. For the *MLH1* non-coding

variant screen the pegRNA-ST cassette was sequenced across all conditions. Additionally, the following amplicons were sequenced from HAP1:PEmax cells without co-selection to validate selective effects for a subset of variants: 164 bp spanning the transcription start site, 138 bp of the 5′-UTR extending into exon 1, 197 bp including exon 11 and adjacent intronic sequence, and 133 bp within intron 15. The same regions were sequenced from control cells (unedited) to quantify baseline sequencing error.

### Variant effect validation experiments

Effects of select variants were reassessed individually or as part of a small pool (i.e., a "minipool").

For *SMARCB1*, 9 variants in total were selected for experimental validation, including 1 expected neutral control (p.Thr335=), 2 expected LoF variants (c.1119−1G>C and p.Pro334*), and 6 variants across coding, intronic and 3′-UTR regions (p.Gln368His, p.Asp369Ala, p.Thr372Ser, c.1119−12C>G, c.*65T>G, and c.*71A>C). For each variant, the pegRNA with the highest activity as defined by ST editing in the large screen was used for the validation experiments.

For *MLH1*, 13 variants were selected, including 2 expected neutral controls (a synonymous variant, c.870G>A, and c.116+220T>G), 2 expected LoF variants (a nonsense variant, c.829G>T, and c.116+1G>A), and 9 variants across upstream, 5′-UTR, coding and intronic regions (c.−7_1del, c.884+3A>G, c.885−3C>G, c.1732−264A>T, c.−219G>T, c.−213G>A, c.−3A>T, c.866A>G, and c.871T>A). For these *MLH1* variants, new pegRNAs were designed using PRIDICT2.[40]

Selected pegRNA-ST cassettes were ordered and synthesized as custom gene fragments (*Twist Bioscience*). Cloning of pegRNAs into lentiviral vectors proceeded as before, except that gene fragments were directly used for Gibson assembly without PCR amplification and the pegRNA scaffold was included in synthesis. Lentivirus particles were produced for individual pegRNA plasmids, as described above, and mixed at equal concentrations to generate the minipool for each gene.

### *SMARCB1* variant effect validation

The procedure for testing the *SMARCB1* minipool remained largely unchanged from the large-scale experiments. In summary, 0.8 × 10^6 HAP1:PE7+MLH1KO were transduced at low MOI (0.1–0.3), followed by selection with puromycin for 3 days. On day 4 post-transduction, cells were split in half with 1 cell pool undergoing co-selection with 5 μM ouabain for 6 days.

For assaying individual pegRNAs in an arrayed format, 0.8 × 10^6 HAP1:MLH1KO cells per pegRNA were transduced at low MOI, followed by puromycin selection for 3 days 0.8 × 10^6 transduced cells were then transfected with pCMV_PE7-2A-BSD using Xfect Transfection Reagent, followed by selection with 5 μg/mL blasticidin for 6 days.

Cell populations were maintained until day 32 following delivery of editing reagents, with splitting and sampling every 2–3 days. Amplification from gDNA and sequencing of endogenous targets and pegRNA-ST cassettes (for minipool experiments only) was performed as described above.

### *MLH1* variant effect validation in HAP1:PE7 cells

Two replicates of 2 × 10^6 HAP1:PE7 cells were transduced with a pegRNA minipool at low MOI (0.1–0.3) and selected with puromycin for 3 days. Co-selection with 5 μM ouabain was performed from day 13 to day 18 post-transduction. On day 19, functional selection with 1.2 μM 6TG was initiated and continued until day 33. Cell pellets were harvested before and after 6TG selection.

To determine splicing consequences of c.884+3A>G and c.885−3C>G, 2 × 10^6 HAP1:PE7 cells were transduced with individual pegRNAs encoding a control variant (c.116+220T>G), c.884+3A>G, or c.885−3C>G, then selected with puromycin for 3 days. On day 7, each cell population was split in two and 6TG selection was performed until harvesting once cells reached confluency (day 22–30).

### *MLH1* variant effect validation in K562:PE7 cells

Two replicates of 0.5 × 10^6 K562:PE7 cells were transduced with the *MLH1* pegRNA minipool at low MOI (0.1–0.3) using spinfection (centrifugation at 1,321 × *g* for 1.5 h at 37°C). On day 2 post-transduction, the media was replaced and the cells were selected with 2 μg/mL puromycin for 5 days. Co-selection with 5 μM ouabain was performed from day 13 to day 18. On day 19, 6TG selection was initiated at 1.2 μM and continued until day 33. Cells were harvested before and after selection. Dead cells were removed prior to cell harvest on day 33 using Dead Cell Removal Kit (#130-090-101, *Miltenyi Biotec*).

### Detection of aberrant splice products

The *SMARCB1* variant c.1119−12C>G and *MLH1* variants c.884+3A>G and c.885−3C>G were selected for mechanistic follow-up to investigate potential changes in splicing. These three variants were experimentally deemed LoF but reported in ClinVar to be of unknown or conflicting clinical significance when screens were initiated in November 2022. As of December 2024, *SMARCB1* c.1119−12C>G and *MLH1* c.885−3C>G remain VUS, whereas c.884+3A>G is now deemed likely pathogenic. To assess these variants' impacts on splicing, total RNA and gDNA were extracted from HAP1 samples to which individual pegRNAs were delivered using the AllPrep DNA/RNA Mini Kit (#80204, *Qiagen*) with QIAShredder columns (#79656, *Qiagen*).

For *SMARCB1*, 100 ng total RNA from day 10 samples corresponding to a neutral control variant (p.Thr335=) and c.1119−12C>G was subjected to RT-qPCR using Luna Universal One-Step RT-qPCR Kit (#E3005S, *NEB*) in technical duplicates with custom primer sets designed using the Primer3 webtool (Table S9). Amplicons obtained from limited-cycle PCR were analyzed by gel electrophoresis using an E-Gel system (#G401002, *Invitrogen*). Amplicons spanning exons 7 to 9 (Ex7-9 amplicons) were prepared for NGS and

sequenced on an Illumina MiSeq. The most frequent sequencing reads were counted and mapped against the reference transcript sequence to identify distinct splicing outcomes between control and c.1119−12C>G samples. Frequencies of splice products were averaged across technical replicates.

For *MLH1*, 2 μg of total RNA extracted from samples corresponding to the post-6TG selection timepoints of the neutral control variant (c.116+220T>G), c.884+3A>G, and c.885−3C>G were reverse-transcribed using SuperScript IV First-Strand Synthesis System and oligo(dT) primers (#18091050, *Thermo Fisher*). Subsequently, the RT product was quantified using the Qubit dsDNA HS kit (#Q33231, *Thermo Fisher*) and 40 ng from each sample was used as input for PCR. Primers were designed to bind within exon 8 and across the exon junction 11–12. Amplicons were visualized via gel electrophoresis and sequenced on an Illumina MiSeq instrument. Paired-end sequencing reads were demultiplexed and merged, and the most frequent reads for each sample were determined and mapped against the reference transcript to quantify the proportion of each splice product.

### Extraction of genomic DNA and PCR amplification

Extraction of gDNA from cell pellets was performed using the DNeasy Blood and Tissue Kit following the supplier's protocol. For amplicon sequencing, genomic sites of interest were amplified by PCR using KAPA HiFi HotStart ReadyMix. Up to 2.5 μg gDNA was used as template per 100 μL PCR volume and reaction mixtures were supplemented with $MgCl_2$ (#AM9530G, *Invitrogen*) to a final concentration of 5 mM. Reactions were supplemented with SYBR Safe (#S33102, *Invitrogen*) and run on a real-time PCR machine to prevent overcycling. Enough reactions were performed and subsequently pooled to maintain at least 1,000-fold average coverage of each pegRNA library when amplifying both pegRNA-ST cassettes and ETs. Primer annealing temperatures for all reactions were predetermined using gradient PCR prior to sample processing. pegRNA-ST amplicons were sequencing-ready after purification of the first PCR from gDNA. ET amplicons were pooled across independent reactions, SPRI-purified and used as template in subsequent PCRs to introduce Illumina sequencing adapters and sample indexes. For the *MLH1* exon 10 experiment, one additional PCR was performed for sample indexing, whereas for the *MLH1* non-coding experiment, an additional nested PCR to install sequencing adapters was performed prior to sample indexing.

### Illumina sequencing

Dual-indexed amplicons with Nextera or TruSeq adapters were sequenced using either an Illumina NextSeq 500 300-cycle kit or an Illumina NovaSeq 6000 SP 300-cycle kit. For deep sequencing of pegRNA-ST cassettes, the length of read 1 was set to 182 nt to capture the full pegRNA sequence and the length of read 2 was set to 118 nt to cover the ST, pegRNA BC and library BC sequences.

## QUANTIFICATION AND STATISTICAL ANALYSIS

All relevant statistical details are listed in figure legends and method details.

### Sequencing data analysis

All bcl files were demultiplexed and converted to fastq files using bcl2fastq2.

### ET analysis

Sequencing reads of ETs from pooled PE screens were processed using DiMSum (--vsearchMinQual 5, --maxSubstitutions 3, otherwise default parameters).[57] Variant counts were further processed with custom Python scripts available via Zenodo (https://doi.org/10.5281/zenodo.14843845). Variants not programmed by pegRNAs within the pool were discarded.

*SMARCB1* ET variant frequencies were corrected for sequencing error by subtraction of background frequencies observed in the negative control sample (non-transduced wildtype cells), with any resulting negative value set to zero. A small number of variants were highly abundant in the negative control sample, likely owing to site-specific sequencing error. Therefore, variants with negative control frequencies above $4 \times 10^{-4}$ were excluded from further analysis. Variant frequencies were averaged across duplicate samples where available.

For *MLH1* ET analyses, log2-ratios of variant frequency on day 20 over day 4 (saturation screen) or day 20 over negative control (non-transduced wildtype cells) (non-coding screen) were calculated. Variants with log2-ratios below 1 were excluded from analysis (i.e., those not enriched over the course of editing). Endogenous function scores were calculated for variants that received a pegRNA-derived function score, as log2-ratios of variant frequency on day 34 over day 20 (post- and pre-6TG selection, respectively), normalized to the median score of synonymous variants (exon 10 screen) or intronic variants annotated as benign in ClinVar (non-coding screen). For the exon 10 screen, normalized endogenous function scores were averaged across conditions to yield a single endogenous function score per variant. These scores were once more normalized to the median synonymous score to produce a final endogenous function score per variant.

### pegRNA-ST read processing

A custom bash script was run on demultiplexed fastq files to extract and write new fastqs for each sequence element, including the protospacer, scaffold, 3′ extension, ST, pegRNA BC, and library BC using cutadapt (version 4.4). Using a custom Python script, sequence elements of each read were queried against a list of expected pegRNA-ST cassettes and reads with non-matching

combinations of elements were discarded. For STs, only the 3′ end of each sequence was used for cross-checking such that reads with edited STs would not be discarded. Matches with up to 10% substitutions per element were allowed (to accommodate sequencing errors), however recombined pegRNA-ST cassettes were recognised and discarded in this step. For each sample, sequencing reads with identical pegRNA identities were tallied to determine pegRNA counts. Next, ST reads were grouped by pegRNA identity and queried for correct editing by searching for a string comprising the intended PE edits flanked by 5 bp on either side. Percentages of correct ST edits were calculated as the number of STs with correct editing over the total number of STs for each pegRNA with at least 10 sequencing reads. pegRNAs with greater than 5% correct ST editing in the negative control sample were excluded from downstream analyses.

### Calculation of pegRNA scores and function scores

For each sample, pegRNA counts were incremented by 1 and converted to frequencies. To measure the change in pegRNA frequency during selection, pegRNA scores were calculated as the log2-ratio of pegRNA frequency in the post-selection sample over pegRNA frequency in the pre-selection sample. To avoid assigning scores to poorly sampled pegRNAs, pegRNAs were filtered on pegRNA frequency in pre-selection samples. Where indicated, pegRNAs were also filtered for editing activity using the percentage of correct ST editing observed for each pegRNA.

For the *SMARCB1* variant screen, pegRNA filtering thresholds were set to $6 \times 10^{-5}$ for pegRNA frequency and 75% correct ST editing for pegRNA activity. For *MLH1* variant screens, pegRNA frequency thresholds were set to $1.4 \times 10^{-4}$ (exon 10 screen) or $1.0 \times 10^{-4}$ (non-coding screen). Activity thresholds were set to 5% correct ST editing for all *MLH1* screens performed in HAP1:PEmax cells and to 25% correct ST editing for all screens performed in HAP1:PEmax+MLH1dn cells. pegRNA scores were normalized to the median pegRNA score of synonymous variants for the *SMARCB1* and *MLH1* saturation mutagenesis screens, or to the median pegRNA score of intronic variants for the *MLH1* non-coding screen.

Function scores for each variant were computed by averaging pegRNA scores for all pegRNAs programming the same variant. For the *SMARCB1* screen, this was achieved by averaging normalized pegRNA scores for each variant. For *MLH1* screens, unnormalized pegRNA scores from each experimental condition were used to determine condition-specific function scores. These were then normalized to the median function score of synonymous variants (exon 10 screen) or to the median function score of intronic variants (non-coding screen) in each condition, prior to averaging function scores across conditions. Function scores averaged across conditions were once more normalized to the median function score of synonymous variants (saturation screen) or intronic variants (non-coding screen), and final function scores were assigned from variants scored in at least two conditions.

To compare function scores derived in HAP1:PEmax and HAP1:PEmax+MLH1dn, cell line-specific scores were calculated. First, condition-specific scores were calculated and normalized as described above. Then, the normalized scores were averaged across both conditions using the same cell line and normalized to the median score of synonymous variants (for the *MLH1* exon 10 screen) or to the median score of intronic variants (for the *MLH1* non-coding screen).

A high-stringency pegRNA activity filter was used to re-analyze the *MLH1* exon 10 screen. This was set to the mean ST editing threshold above which the range of pLoF (i.e., nonsense and canonical splice) variant scores is non-overlapping with the range of pNeut (i.e., synonymous) variant scores, corresponding to 36% mean ST editing per variant.

To classify variants in each screen as LoF, a normal distribution of neutral function scores was modeled from synonymous variants (*SMARCB1* and *MLH1* exon 10 screens) or intronic variants (*MLH1* non-coding screen) to calculate *p*-values for each function score. To correct for multiple hypothesis testing, the Benjamini-Hochberg (BH) procedure was applied. Variants with q-values less than 0.05 for the *SMARCB1* screen and less than 0.01 for the *MLH1* screen were deemed LoF variants.

### AUC analysis

AUC measurements for separating pLoF and pNeut variants by function score were computed over a continuous range of ST editing thresholds. At each ST editing threshold sampled, function scores were re-calculated from the set of pegRNAs with ST editing percentages above threshold. For the *MLH1* exon 10 screen, synonymous variants were defined as pNeut and nonsense and canonical splice variants as pLoF. For the *MLH1* non-coding screen, intronic variants were defined as pNeut and canonical splice variants as pLoF.

To define a high-stringency ST editing threshold for the exon 10 screen, AUC measurements for distinguishing LoF variants by function score were computed over a continuous range of mean ST editing thresholds. Mean ST editing rates were calculated by averaging the editing rates of pegRNAs programming the same variant across conditions. The high-stringency analysis excluded pegRNAs with ST editing rates less than 5% in PEmax cell lines or less than 25% in PEmax-MLH1dn cell lines, as these were removed prior.

### Comparing function scores with orthologous datasets

The crystal structure of human MLH1 (PDB: 4P7A) was imported to PyMol. Maximum function scores for missense variants tested at each amino acid position were calculated and used to color-code residues. Negative function scores were set to 0.0 for this analysis.

*MLH1* variant pathogenicity assertions were retrieved from ClinVar in August 2023. For benchmarking of function scores, variants deemed "pathogenic" or "likely pathogenic" were grouped together, as were variants deemed "benign" or "likely benign". Variants deemed to be of "uncertain significance" or with "conflicting interpretations of pathogenicity" were also grouped together as VUS.

CADD scores (v1.6)[60] and annotations were obtained for all *SMARCB1* and *MLH1* SNVs (https://cadd.gs.washington.edu/download), inclusive of SpliceAI scores. A single maximum SpliceAI score was determined for each variant from independent scores for "acceptor gain", "acceptor loss", "donor gain", and "donor loss" scores and used for comparison to function scores. Variant "consequence" annotations for MNVs, deletions, and insertions were manually curated. The *MLH1* variant c.−7_1del was annotated as a 5′-UTR variant throughout analyses except where explicitly indicated "start loss".

### Analysis of sequencing data from validation experiments

For *SMARCB1*, the pegRNA-ST data was processed as before using custom scripts to derive ST editing rates and log2-fold-changes of pegRNA frequencies at day 32 over day 10, which were normalized to the neutral control variant (p.Thr335=). DiMSum was used to determine variant frequencies at endogenous targets in the minipool experiment. Variant frequencies in experiments where pegRNAs were tested individually were determined using CRISPResso2, and variant scores were normalized to wildtype frequency fold-change.

For *MLH1*, pegRNA-ST data preprocessing, derivation of correct ST-editing rates and calculation of function scores from sequencing data was performed as described above. c.1732−264A>T was excluded from analysis due to low editing activity of the newly designed pegRNA (less than 5%). Replicate-specific function scores were normalized to the median function score of the two neutral controls. Final function scores (averaged across replicates) were likewise normalized to the median of neutral controls. *MLH1* ET editing rates were determined using DiMSum, as described in the section "ET analysis". c.116+1G>A was excluded from ET analysis due to the absence of a suitable primer pair for gDNA amplification. Endogenous function scores were calculated as described above, with replicate-specific endogenous function scores normalized to the median score of neutral controls, prior to averaging across replicates and once more normalizing to the neutral median score to derive final endogenous function scores.

