## [Document S2. Transparent peer review record · Cell Genomics]

High-throughput screening of human genetic variants by pooled prime editing

Michael Herger, Christina M. Kajba, Megan Buckley, Ana Cunha, Molly Strom, and Gregory M. Findlay

Summary

Initial submission: Received : Mar 22, 2024

Scientific editor: Sara Rohban

First round of review: Number of reviewers: 2
Revision invited : Apr 15, 2024
Revision received : Jan 10, 2025

Second round of review: Number of reviewers: 2
Accepted : Feb 20, 2025

Data freely available: YES

Code freely available: YES

This transparent peer review record is not systematically proofread, type-set, or edited. Special characters, formatting, and equations may fail to render properly. Standard procedural text within the editor's letters has been deleted for the sake of brevity, but all official correspondence specific to the manuscript has been preserved.

Referees' reports, first round of review

Reviewer #1:

In the present manuscript titled "High-throughput screening of human genetic variants by pooled prime editing", the authors successfully introduced a pooled prime editing (PE) platform in haploid human cells (HAP1) to efficiently analyze variants in endogenous context, through exploiting an ATP1A1 T804N mutation dependent co-selection strategy. And they characterized the platform in the context of negative selection by testing efficiencies of over 7,500 pegRNAs targeting SMARCB1 for installing loss-of-function variants. Then, they further assessed the potential function of the coding and non-coding MLH1 variants, and identified pathogenic variants acting via multiple mechanisms with high specificity. In general, the authors combined co-selection strategy with prime editing screening to develop a novel prime editing (PE) platform in haploid human cells, and further facilitated both positive and negative selection screens by filtering for highly active pegRNAs. This work is well-designed and interesting. Nevertheless, I would like to see the following points addressed in the manuscript before its publication in Cell Genomics.

Comments/Suggestions

1. In line 72-74, the author questioned that "at present it remains unclear to what extent large PE screens are feasible" in the introduction, but this work seems to be no solution to this problem, and authors also didn't construct a pegRNA library with sufficient throughput, please discuss.
2. In line 92-96, the author claimed "Since HAP1 cells are mismatch repair (MMR) proficient, a second HAP1 line was generated with concomitant expression of a dominant negative MLH1 protein (MLH1dn) shown to enhance prime editing efficiency". However, no direct experiment result showed HAP1:PEmax+MLH1dn performed higher efficiency than HAP1:PEmax. To verify it at endogenous sites is necessary.

3. In figure 1, authors introduced an ATP1A1 T804N mutation dependent co-selection strategy to improve efficiency. Co-selection systems enable the enrichment of intended edits by co-editing the ATP1A1 T804N mutation, leading to a selectable phenotype. The results of figure 1D only proved that ouabain selection can enrich cells with T804N mutation, but endogenous editing efficiency was not indicated. Please verify it in HAP1:PEmax+MLH1dn and HAP1:PEmax cell lines according to the design of figure 1C.

4. Authors found that stabilized scaffolds outperformed the original in figure 1E, please give the corresponding sequence and structural information of all stabilized scaffolds.

5. To assess whether surrogate target (ST) editing rates can accurately reflect endogenous target (ET) editing rates, authors directly sequenced ETs in SMARCB1 amplified from the same cells used to measure pegRNA frequencies and ST editing. But the ET variant frequency of intended edits was very low, only about 1×10^{-4} in figure S4, indicating that sequencing errors will have a big impact on the analysis results. Thus, the ST variant frequency may only perform a general correlation with ET variant frequency ($r=0.53$), please discuss.

6. In figure 3, authors assessed MLH1 variants via 6-thioguanine selection, and distinguished LoF variants with high accuracy. What's more. In figure 4, authors also assayed non-coding MLH1 variants, and identified pathogenic variants acting via multiple mechanisms with high specificity. I suggested authors should perform appropriate functional validation based on screening results to further demonstrate the advantage of developed screening platform.

7. Due to HAP1:PEmax+MLH1dn cells with expression of a dominant negative MLH1 protein (MLH1dn protein), therefore, it was unreasonable to use the HAP1:PEmax+MLH1dn cells for MLH1 variants screening, please give a detailed explanation.

Reviewer #2:

Summary:

Many human diseases have a strong germline or somatic genetic component. The (mutant) genes causally involved in a number of these diseases are known; however, the key challenge of connecting disease-associated genetic variants to function and mechanism in the context of human disease remains daunting. Moreover, how those variants interact functionally with the diversity of genomes that exist in human population is an even bigger challenge. Put simply, disease-associated variant-to-function studies are urgently needed in order to understand, predict, and perhaps even treat/correct genetic diseases. This is the main challenge that Herger et al tackle in this study.

The scale, precision, and throughput of methods designed to perform variant-to-function studies have significantly evolved over the last decade or so thanks to CRISPR-Cas technologies. While this is not a review of the literature, it is worth noting that Greg Findlay has been a pioneer in this space for some time now and his work has motivated a number of follow-up studies and technology development efforts in the field. Most notably, Greg Findlay, Jay Shendure, and colleagues demonstrated early on that CRISPR-Cas nuclease technologies could be leveraged to perform Saturation Genome Editing (SGE) in mammalian cells to functionally interrogate thousands of endogenous genetic variants in a multiplexed fashion (e.g. PMID: 25141179, PMID: 30209399). Although all of these studies were performed in HAP1 haploid mammalian cells (including the one being reviewed here) in order to maximize editing efficiencies and simplify variant-to-function measurements, these efforts continue to highlight the power of SGE to functionally interrogate genetic variants in a massively parallel fashion.

These strategies have always had at least four shortcomings: 1) HDR remains inefficient; thus, editing two or more endogenous alleles of the same gene is very, very challenging; 2) HDR is difficult to scale up to perform comprehensive pooled SGE because each edit requires a separate donor template, and each section of the gene therefore usually requires different HDR "pools"; 3) cutting often generates indels, potentially obscuring the true effects of endogenous genetic variants; and 4) mammalian cells are diploid, and many disease variants are observed at homozygosity or (may) exhibit important genetic interactions with the wild type or otherwise non-variant allele (e.g. LOH events in cancer). Herger et al tackle the first three challenges in

this manuscript using a relatively new technology called CRISPR prime editing.

Prime editing enables the generation of any class of small genetic variants at defined genetic loci. With this capability, prime editing can be applied to evaluate the effects of genetic variants in their endogenous genomic context. To date, only a small number of studies have applied prime editing in a high-throughput, pooled format to interrogate hundreds to thousands of genetic variants in parallel. This is due in part to the relatively low efficiency of prime editing relative to other genome editing techniques, and the difficulty in predicting efficient pegRNA designs that can engineer a given variant.

With these challenges in mind, Herger et al explore the optimal parameters of a pooled prime editing screen, and use these optimized parameters to apply pooled prime editing to evaluate (1) SMARCB1 variants, (2) MLH1 coding variants, and (3) MLH1 splice site and intronic variants. They show that using ouabain co-selection, a well-described method to increase the frequency of genome edited cells (including Cas9 nuclease-mediated NHEJ and HDR, e.g. PMID: 28417998, and CRISPR prime editing, e.g. PMID: 36207338) improves the practical efficiency of prime editing by selecting for cells with active prime editors. In addition, they show that designing multiple ($n=9$) pegRNAs for a given variant and using "surrogate target" (ST) sites to obtain an integrated readout of pegRNA editing activity and subsequently filter out low efficiency pegRNAs allows for improved discrimination between putative loss of function and neutral variants. They further define editing thresholds for their particular selection conditions that result in high discrimination between these putative loss of function and neutral variants.

As stated above, the development and implementation of strategies to apply high-throughput prime editing for massively parallel variant-to-function characterization studies remains a very active area of investigation that is still in its infancy. While this field is currently undergoing an explosion, the study by Herger et al undoubtedly represents one of the earliest attempts to optimize and deploy high-throughput prime editing, and they nicely demonstrate the power of integrating co-selection, ST sites, and careful computational analysis of the resulting data in haploid mammalian cells to functionally interrogate endogenously engineered variants associated with human genetic diseases. The strategies described in this study should therefore be immediately applicable to many other efforts aimed at investigating the functional consequences of disease-associated variants, but also to perform SGE studies aimed at native gene function studies. As such, I genuinely believe that a revised manuscript of this study

should be considered for publication in Cell Genomics. Below are my specific comments and recommendations for manuscript revision.

General Comments:

(1) Unless I missed it, the authors do not perform any functional experiments to validate the results of their screen, e.g. through individual testing of pegRNAs. Instead, they rely on computational variant effect predictors to validate their findings. While their predictive analysis is important and informative, it is virtually impossible to state with any significant certainty that the effects and behavior of the mutations they engineer and interrogate in their HAP1 system are biologically impactful and meaningful. It is thus critical that they perform experiments with individual pegRNAs and analyze the biological consequences of these perturbations, ideally in HAP1 cells but also in at least one diploid mammalian counterpart to make it clear that their methods are not strictly limited to haploid mammalian cells.

(2) Secondly, the authors frame the use of ST sites to filter out low efficiency pegRNAs and improve screening resolution as a fundamentally novel aspect of their approach. However, a recent publication in Nature Biotechnology, which they cite as a bioRxiv pre-print, and whose computational tool (PEGG) they used to generate their pegRNA libraries, performed this exact approach (PMID: 38472508).

(3) The ouabain co-selection approach is quite powerful but it has already been shown before to boost the selection of cells with functional prime editing activity (PMID: 36207338).

(4) While technically impressive, their screening data does not seem to provide additional biological insights beyond those already offered by variant effect prediction tools.

I believe points (1) and (4) above are important and could be boosted by the suggestions stated in (1) and below. Points (2) and (3) have to do with the novelty of the approach itself in the context of the current literature, including not referencing a published contemporary peer-reviewed study that used the same "ST" strategy described in point (2).

Major Comments:

(1) The authors describe in the discussion (Lines 451-460):

"In the context of well-established MAVEs for assessing variants by genome editing, such as base editing screens, saturation genome editing via HDR, and saturation prime editing, our platform offers the advantage of being able to install any set of short variants virtually anywhere in the genome, provided an active pegRNA can be designed.

Current limitations stem from the fact that only a limited fraction of pegRNAs lead to robust editing. This explains the incomplete scoring of variants for which we designed pegRNAs. As we show for variants in both SMARCB1 and MLH1, separation of signal from noise depends strongly on discerning more active pegRNAs, as editing percentages vary highly. In our platform, this is made possible via inclusion of STs with each pegRNA assayed."

The framing of the use of ST sites and the subsequent filtration of low efficiency pegRNAs as a fundamentally novel technological advancement seems inaccurate and should be revised. Specifically, a recent paper, whose bioRxiv preprint is cited by the authors and whose computational tool was used to generate the pegRNA libraries used in this study, described this exact methodology (PMID: 38472508). Given this contemporary study was peer-reviewed and published before Herger et al submitted their study to Cell Genomics (and to bioRxiv in parallel), their narrative should be modified to reflect the previous utilization of this method in the context of pooled prime editing. It seems dishonest to ignore this.

(2) In the three screens that were performed (SMARCB1, MLH1 coding, MLH1 noncoding) there is no validation of individual pegRNAs. The authors primarily make use of variant effect predictors (CADD score, ClinVar annotation, SpliceAI) to "validate" the results of their screen. Validation of individual pegRNAs, including top hits that are not predicted to have a strong effect based on computational tools, would significantly add to this manuscript, and potentially provide novel biological insights. It is thus important that the authors perform experiments with individual pegRNAs and analyze the biological consequences of these perturbations, ideally in HAP1 cells but also in at least one diploid mammalian counterpart to illustrate that their methods are not strictly limited to haploid mammalian cells. This is important because readers may think that this technology is only useful at scale if one has to edit a single allele of a gene. However, a number

of studies have shown that this is not the case for a handful of loci. In this regard, the amplicon sequencing of endogenous loci that is performed by the authors is insufficient to validate that the effects they show validate in a non-pooled format and produce biologically meaningful results.

(3) The Github link provided is not publicly accessible and thus cannot be reviewed.

(4) Supplementary Figure 8 shows that there are differences in the pegRNA function score between the PEmax and PEmax + MLH1dn conditions. The authors fail to address these differences, which are particularly relevant given that MLH1 is the gene being assessed, and presumably the MLH1dn protein could functionally and/or directly interact with MLH1 and modify any variant functions.

Minor Comments:

In the first part of the manuscript, the authors conclude that MNVs are less efficiently installed than SNVs. However, the analysis provided in Figure 1H shows that a subset of MNVs are engineered much more efficiently than SNVs. Are there sequence-specific effects at play here? Please discuss.

As mentioned above, have the authors performed any experiments in non-haploid cell types? How does the efficiency of the oubain co-selection method compare if so?

A comparison of the features that are predictive of correct editing (Figure 2E) to other works in the space (PRIDICT, DeepPrime) would be useful to understand whether these results are biased by the pegRNA design methods employed by the authors.

Given that these features do not appear to match up with these previous papers, a further explanation or exploration would be useful here both for the reader but also to lend more credential to the current study.

Authors' response to the first round of review

Reviewer comments are reproduced in black.

Author responses are in blue.

Quoted manuscript changes are in green.

We thank both reviewers for their very helpful comments. We have addressed all points in extensively revising the manuscript, and detail all changes in response to specific reviewer comments.

Reviewers' Comments:

Reviewer #1:

In the present manuscript titled "High-throughput screening of human genetic variants by pooled prime editing", the authors successfully introduced a pooled prime editing (PE) platform in haploid human cells (HAP1) to efficiently analyze variants in endogenous context, through exploiting an ATP1A1 T804N mutation dependent co-selection strategy. And they characterized the platform in the context of negative selection by testing efficiencies of over 7,500 pegRNAs targeting SMARCB1 for installing loss-of-function variants. Then, they further assessed the potential function of the coding and non-coding MLH1 variants, and identified pathogenic variants acting via multiple mechanisms with high specificity. In general, the authors combined co-selection strategy with prime editing screening to develop a novel prime editing (PE) platform in haploid human cells, and further facilitated both positive and negative selection screens by filtering for highly active pegRNAs. This work is well-designed and interesting. Nevertheless, I would like to see the following points addressed in the manuscript before its publication in Cell Genomics.

We thank the reviewer for these positive comments.

Comments/Suggestions

1. In line 72-74, the author questioned that "at present it remains unclear to what extent large PE screens are feasible" in the introduction, but this work seems to be no solution to this problem, and authors also didn't construct a pegRNA library with sufficient throughput, please discuss.

We appreciate the reviewer's point that our proof-of-concept experiments feature more modestly sized libraries compared to what has been demonstrated with established technologies. Given the lower editing rates associated with PE, we felt using smaller pegRNA libraries and representing individual edits with multiple pegRNAs was necessary to increase our chance of including highly active pegRNAs. Additionally, it was necessary to perform each of these initial screens in multiple conditions to establish best practices going forward, which limited our ability to scale.

In **Discussion**, we outline several recommendations to boost PE efficiency, which will inevitably allow larger screens to be performed in the future. As suggested by the reviewer, we have now modified this section to highlight the modest size of our current libraries and to make clear that future improvements in editing efficiency will allow larger libraries to be assayed:

Importantly, as PE reagents continue to improve and pegRNA design tools mature, we anticipate a larger fraction of pegRNAs in future experiments will produce accurate data. In this study, we prioritized testing multiple modestly sized pegRNA libraries across different screening conditions and functional assays. However, we anticipate assaying substantially larger pegRNA libraries will be readily achievable with our platform, as has been demonstrated for other PE platforms²⁹. Based on this work, specific recommendations for performing future PE screens in HAP1 include: 1.) allowing ample time for edits to accrue, 2.) using a stabilized pegRNA scaffold, such as F+E v1, and 3.) including STs to filter out inactive pegRNAs. Though we saw a consistent benefit of ouabain co-selection, increases in editing rates were modest, suggesting this optimization may not always be necessary depending on the application. Implementing recently described pegRNA scoring tools^{36,37} in experimental design promises to improve coverage and reduce the number of pegRNAs with low editing activities. Furthermore, newer prime editors, such as PE7²⁴, may further improve editing rates and screening data quality.

2. In line 92-96, the author claimed "Since HAP1 cells are mismatch repair (MMR) proficient, a second HAP1 line was generated with concomitant expression of a dominant negative MLH1 protein (MLH1dn) shown to enhance prime editing efficiency". However, no direct experiment result showed HAP1:PEmax+MLH1dn performed higher efficiency than HAP1:PEmax. To verify it at endogenous sites is necessary.

We agree that verifying increased editing activity in HAP1:PEmax+MLH1dn is necessary. Indeed, we looked at this directly by sequencing the endogenous loci of cells edited in our *MLH1* screens. **Figure S7B** (using updated figure numbering) shows variant frequencies measured at endogenous targets (ETs) on day 20, both with and without ouabain co-selection, reproduced here:

To make this essential comparison more clear, we have edited this sentence in the manuscript to directly report that endogenous editing rates were higher in HAP1:PEmax+MLH1dn:

We first compared ST editing and ET editing across cell lines and co-selection strategies. We observed higher editing at both STs and ETs in the HAP1:PEmax+MLH1dn cell line compared to HAP1:PEmax (**Figure S7B**). The median variant frequency was 1.5-fold higher in HAP1:PEmax+MLH1dn compared to HAP1:PEmax in cells without co-selection, and 1.9-fold higher in cells in which co-selection was performed.

3. In figure 1, authors introduced an ATP1A1 T804N mutation dependent co-selection strategy to improve efficiency. Co-selection systems enable the enrichment of intended edits by co-editing the ATP1A1 T804N mutation, leading to a selectable phenotype. The results of figure 1D only proved that ouabain selection can enrich cells with T804N mutation, but endogenous editing efficiency was not indicated. Please verify it in HAP1:PEmax+MLH1dn and HAP1:PEmax cell lines according to the design of figure 1C.

Establishing whether the co-selection strategy leads to increased editing efficiency in the context of PE screening is important, and we agree this is not established in Figure 1D.

However, we do show this result in **Figure 2F** for surrogate target (ST) editing, and in **Figures S5B and S5C** for endogenous target (ET) editing (reproduced below). These figures reflect editing rates across many independent target sites sampled at different timepoints, establishing a consistent increase in intended editing with co-selection.

4. Authors found that stabilized scaffolds outperformed the original in figure 1E, please give the corresponding sequence and structural information of all stabilized scaffolds.

Thanks for this suggestion. All scaffold sequences are included in **Table S9**. We have added new figures showing the alignment and structural information for each scaffold design (**Figures S1B-F**):

Figure S1. pegRNA optimization for *ATP1A1*-T804N edit in HEK293T and pegRNA scaffold designs.

...(B) Sequence alignment of pegRNA scaffolds used in this study for optimizing PE efficiency. Base changes against the original scaffold are colored red.

(C-F) The predicted secondary structure of each scaffold is shown. pegRNA-specific elements are colored blue.

5. To assess whether surrogate target (ST) editing rates can accurately reflect endogenous target (ET) editing rates, authors directly sequenced ETs in *SMARCB1* amplified from the same cells used to measure pegRNA frequencies and ST editing. But the ET variant frequency of intended edits was very low, only about 1×10^{-4} in figure S4, indicating that sequencing errors will have a big impact on the analysis results. Thus, the ST variant frequency may only perform a general correlation with ET variant frequency ($r=0.53$), please discuss.

This is a great point and something we considered carefully in our analysis.

Firstly, we performed deep sequencing of unedited ETs to assess the contribution of PCR and sequencing error to our observed ET variant frequencies. This allowed us to perform a background correction when calculating ET editing rates for *SMARCB1* variants, as described in **Methods**. Therefore, it is highly unlikely that the reported correlation of $r = 0.53$ is driven by sequencing error.

Indeed, if we restrict this analysis to only multi-nucleotide variants (MNVs) and deletions (DELs) the correlation improves. Whereas the overall correlation for *SMARCB1* variants

in the co-selected D10 sample was $r = 0.53$, this can be split by type of edit: SNVs only, $r = 0.53$; MNVs only, $r = 0.58$; DELs only, $r = 0.68$. The fact that MNVs and DELs, which are not commonly impacted by sequencing error in the way that SNVs are, display a similar but somewhat higher correlation indicates sequencing error is not causing this result.

One limitation to this analysis is that many pegRNAs encode the same variant. We calculated a variant-level ST editing rate using the sum of all individual pegRNA ST edits covering the same variant, and compared this to the observed ET editing rate for the variant. To the extent that such weighted averaging introduces discordance, the correlation between ST and ET editing would be higher if each variant was encoded by a single pegRNA.

Critically, we ultimately show that using an ST editing rate filter dramatically improves the quality of the data. This would not work if there were no correlation between ST editing rate and ET editing rate. We note that a high correspondence between ST editing and ET editing has also now been reported elsewhere, work that we highlight in our modified **Discussion** as follows:

The improvement in data quality observed upon filtering pegRNAs by ST editing reflects the fact that ST editing is a reasonable proxy for ET editing, as we have shown (**Figures 2G and S5A**). Notably, the same approach of using STs (i.e., “sensors”) to filter inactive pegRNAs from analysis was also shown to improve PE screening results in a recent study of *TP53* variants²⁸.

6. In figure 3, authors assessed MLH1 variants via 6-thioguanine selection, and distinguished LoF variants with high accuracy. What's more. In figure 4, authors also assayed non-coding MLH1 variants, and identified pathogenic variants acting via multiple mechanisms with high specificity. I suggested authors should perform appropriate functional validation based on screening results to further demonstrate the advantage of developed screening platform.

We agree with the need to functionally validate screen hits to establish the method's advantages and have now completed multiple experiments to address this point also raised by Reviewer 2. In summary, we have:

1. Tested a minipool of pegRNAs encoding depleted *SMARCB1* variants and validated the correct installment of edits and selection against LoF alleles. We also

demonstrate that a newly identified intronic LoF variant leads to aberrant splicing by testing it individually.

2. Tested a minipool of newly designed pegRNAs encoding neutral and LoF *MLH1* variants and validated selection for LoF alleles upon 6TG treatment. Importantly,

we also tested the same pool in K562 cells and show effects to be highly concordant between HAP1 and K562.

3. Demonstrated the molecular effects of two clinically relevant *MLH1* variants deemed LoF by introducing them individually to cells and observing aberrant splicing.

These results broadly confirm the assay's utility as a tool for discovery, yielding functional insights not clear from computational predictors alone while also shining light on some potential sources of noise in the screens, as to be expected in light of the imperfect separation of control variants. We describe these extensive new experiments in full below (please see section "**Detailed description of validation experiments**" on p.27), and have added 2 supplemental figures, several paragraphs of **Results**, and complete descriptions of the work in **Methods**.

7. Due to HAP1:PEmax+MLH1dn cells with expression of a dominant negative MLH1 protein (MLH1dn protein), therefore, it was unreasonable to use the HAP1:PEmax+MLH1dn cells for MLH1 variants screening, please give a detailed explanation.

The reviewer raises a good point. The choice of whether or not to use the PEmax+MLH1dn HAP1 line for *MLH1* screening was something we thought about extensively but failed to address adequately before. Now, we more thoroughly detail our choice to perform screens in both HAP1:PEmax and HAP1:PEmax+MLH1dn, the difference in screen performance across the two lines, and our decision to combine data from experiments performed in both lines to produce final scores (while also making scores from individual experiments available).

We reasoned that expression of MLH1dn would increase editing rates, but we shared the concern that this might compromise the 6TG read-out. This is why we first asked whether stable MLH1dn expression in HAP1 confers the same degree of 6TG resistance as *MLH1* LoF mutations. The results of our 6TG titration (now **Figure S7A**) indicate MLH1dn

expression does not confer the same degree of 6TG resistance as knockout of *MLH1*, but only partial resistance:

Figure S7. Effects of 6TG on growth and editing rates compared across HAP1 lines with modified MMR function.

(A) Viable cell counts are plotted by 6TG dose following 6 days treatment for HAP1:PEmax, HAP1:PEmax+MLH1dn, and HAP1:PE2+MLH1KO.

This led us to reason that LoF *MLH1* mutations introduced in HAP1:PEmax+MLH1dn would be enriched over neutral variants in our screens, as was observed. However, not knowing which line would perform optimally for screening, we still performed the *MLH1* screens in both HAP1:PEmax and HAP1:PEmax+MLH1dn for comparison, using a slightly higher dose of 6TG to perform selection in HAP1:PEmax+MLH1dn (detailed in **Methods**).

Our analyses indicate that the HAP1:PEmax line ended up performing best overall, as shown with the AUC analysis in **Figure 3B**. Yet, LoF variants were also clearly enriched in the HAP1:PEmax+MLH1dn line. One reason that the HAP1:PEmax+MLH1dn line still performed reasonably well was that editing rates were higher, as shown in **Figure S7B**. Given the reasonable function score correlations between cell lines (**Figures S9C and S9F**) and the expectation that there'll be some noise in each screen, we decided to produce final scores by averaging across lines (see **Methods**). Importantly, scores obtained in each individual line (HAP1:PEmax and HAP1:PEmax+MLH1dn) are also made available in **Tables S6 and S8**.

We have also now performed a more careful analysis of the data derived in HAP1:PEmax+MLH1dn to ask if it adds predictive power to the overall score set. Indeed, we observe a higher correlation to orthologous metrics of variant effect such as CADD and SpliceAI scores when data from both cell lines are averaged to produce final scores (compared to using data from either line alone). We now include this analysis in **Figures S10A-C**. The improvement in correlation when averaging scores across lines makes sense considering these proof-of-concept screens display some noise, such that pooling data from more independent experiments is bound to improve accuracy.

The reviewer is right to point out that certain variants score differently between the two lines, but this is only a small subset of variants assayed. To enable more transparent comparison of scores between HAP1:PEmax and HAP1:PEmax+MLH1dn, we have added a heatmap showing scores of loss-of-function variants across the four experimental conditions (**Figures S10D and S10E**). Overall, big differences between cell lines are not common. As MLH1dn expression alone leads to lower 6TG sensitivity, we cannot exclude the possibility that variants scoring as LoF only when MLH1dn is expressed partially impair function.

To confirm whether the observed differences between lines were likely to be real, we chose 2 missense variants that scored highly in HAP1:PEmax+MLH1dn but not HAP1:PEmax to retest with new pegRNA designs in PE7-expressing cells without MLH1dn expression (**Figures S14C and S14D**). These variants were once more not enriched in cells lacking MLH1dn expression, suggesting expression of MLH1dn may contribute to the differences between lines.

On the whole, these new data indicate MLH1dn expression may impact screening results for a small subset of variants. As our focus is on establishing proof-of-concept for a new variant screening platform, we feel further investigation of the molecular basis of differences between cell lines with and without MLH1dn expression is beyond scope. Thinking forward, variants to be assayed in genes outside the MMR pathway are unlikely to be impacted by MLH1dn expression.

Figure S10. Consistency of *MLH1* variant effects between HAP1:PEmax and HAP1:PEmax+*MLH1dn*.

(A-C) Correlation between function scores and CADD scores (top), and between function scores and SpliceAI scores (bottom) using data generated in HAP1:PEmax only (A), in HAP1:PEmax+*MLH1dn* only (B), or by combining data from both HAP1:PEmax and HAP1:PEmax+*MLH1dn* (C). Pearson correlation coefficients (r) are shown for each comparison. (D-E) Heatmaps of function scores across the four experimental conditions for all variants deemed LoF in the exon 10 screen (D) and for all variants deemed LoF in the non-coding screen (E) are shown. The color scale corresponds to the minimum and maximum score observed in any condition.

To make our scoring choice more transparent, we've modified our **Results** to explicitly report that optimal scoring is achieved when data from HAP1:PEmax and HAP1:PEmax+*MLH1dn* are combined in analysis, and that only a small subset of variants scored discordantly across lines.

Considering the possibility that MLH1dn expression may impact the effects of *MLH1* variants, we also computed cell-line specific function scores using data from only HAP1:PEmax or only HAP1:PEmax+MLH1dn. However, correlations to variant effect predictors were strongest when data from both cell lines were combined (**Figures S10A-C**). Though rare, a small subset of LoF variants displayed stronger effects with MLH1dn expression (**Figures S10D and S10E**), for example, c.866A>G and c.871T>A.

Additionally, we have modified the **Discussion** to summarize the trade-off between achieving optimal editing rates and producing the highest quality scores for variants in *MLH1*:

Whereas our data suggest that a small subset of *MLH1* variants may be influenced by expression of MLH1dn, for assaying variants in genes outside the MMR pathway, MLH1dn expression or *MLH1* knockout will likely increase editing without confounding interpretation.

Reviewer #2:

Summary:

Many human diseases have a strong germline or somatic genetic component. The (mutant) genes causally involved in a number of these diseases are known; however, the key challenge of connecting disease-associated genetic variants to function and mechanism in the context of human disease remains daunting. Moreover, how those variants interact functionally with the diversity of genomes that exist in human population is an even bigger challenge. Put simply, disease-associated variant-to-function studies are urgently needed in order to understand, predict, and perhaps even treat/correct genetic diseases. This is the main challenge that Herger et al tackle in this study.

The scale, precision, and throughput of methods designed to perform variant-to-function studies have significantly evolved over the last decade or so thanks to CRISPR-Cas technologies. While this is not a review of the literature, it is worth noting that Greg Findlay has been a pioneer in this space for some time now and his work has motivated a number of follow-up studies and technology development efforts in the field. Most notably, Greg Findlay, Jay Shendure, and colleagues demonstrated early on that CRISPR-Cas nuclease technologies could be leveraged to perform Saturation Genome Editing (SGE) in mammalian cells to functionally interrogate thousands of endogenous genetic variants in a multiplexed fashion (e.g. PMID: 25141179, PMID: 30209399).

Although all of these studies were performed in HAP1 haploid mammalian cells (including the one being reviewed here) in order to maximize editing efficiencies and simplify variant-to-function measurements, these efforts continue to highlight the power of SGE to functionally interrogate genetic variants in a massively parallel fashion.

These strategies have always had at least four shortcomings: 1) HDR remains inefficient; thus, editing two or more endogenous alleles of the same gene is very, very challenging; 2) HDR is difficult to scale up to perform comprehensive pooled SGE because each edit requires a separate donor template, and each section of the gene therefore usually requires different HDR "pools"; 3) cutting often generates indels, potentially obscuring the true effects of endogenous genetic variants; and 4) mammalian cells are diploid, and many disease variants are observed at homozygosity or (may) exhibit important genetic interactions with the wild type or otherwise non-variant allele (e.g. LOH events in cancer). Herger et al tackle the first three challenges in this manuscript using a relatively new technology called CRISPR prime editing.

Prime editing enables the generation of any class of small genetic variants at defined genetic loci. With this capability, prime editing can be applied to evaluate the effects of genetic variants in their endogenous genomic context. To date, only a small number of studies have applied prime editing in a high-throughput, pooled format to interrogate hundreds to thousands of genetic variants in parallel. This is due in part to the relatively low efficiency of prime editing relative to other genome editing techniques, and the difficulty in predicting efficient pegRNA designs that can engineer a given variant.

With these challenges in mind, Herger et al explore the optimal parameters of a pooled prime editing screen, and use these optimized parameters to apply pooled prime editing to evaluate (1) SMARCB1 variants, (2) MLH1 coding variants, and (3) MLH1 splice site and intronic variants. They show that using ouabain co-selection, a well-described method to increase the frequency of genome edited cells (including Cas9 nuclease-mediated NHEJ and HDR, e.g. PMID: 28417998, and CRISPR prime editing, e.g. PMID: 36207338) improves the practical efficiency of prime editing by selecting for cells with active prime editors. In addition, they show that designing multiple (n=9) pegRNAs for a given variant and using "surrogate target" (ST) sites to obtain an integrated readout of pegRNA editing activity and subsequently filter out low efficiency pegRNAs allows for improved discrimination between putative loss of function and neutral variants. They further define editing thresholds for their particular selection conditions that result in high discrimination between these putative loss of function and neutral variants.

As stated above, the development and implementation of strategies to apply high-throughput prime editing for massively parallel variant-to-function characterization studies remains a very active area of investigation that is still in its infancy. While this field is currently undergoing an explosion, the study by Herger et al undoubtedly represents one of the earliest attempts to optimize and deploy high-throughput prime editing, and they nicely demonstrate the power of integrating co-selection, ST sites, and careful computational analysis of the resulting data in haploid mammalian cells to functionally interrogate endogenously engineered variants associated with human genetic diseases. The strategies described in this study should therefore be immediately applicable to many other efforts aimed at investigating the functional consequences of disease-associated variants, but also to perform SGE studies aimed at native gene function studies. As such, I genuinely believe that a revised manuscript of this study should be considered for publication in Cell Genomics. Below are my specific comments and recommendations for manuscript revision.

We thank the reviewer for the positive comments.

General Comments:

(1) Unless I missed it, the authors do not perform any functional experiments to validate the results of their screen, e.g. through individual testing of pegRNAs. Instead, they rely on computational variant effect predictors to validate their findings. While their predictive analysis is important and informative, it is virtually impossible to state with any significant certainty that the effects and behavior of the mutations they engineer and interrogate in their HAP1 system are biologically impactful and meaningful. It is thus critical that they perform experiments with individual pegRNAs and analyze the biological consequences of these perturbations, ideally in HAP1 cells but also in at least one diploid mammalian counterpart to make it clear that their methods are not strictly limited to haploid mammalian cells.

We fully agree and have now done as the reviewer suggested. We discuss this in-depth below.

(2) Secondly, the authors frame the use of ST sites to filter out low efficiency pegRNAs and improve screening resolution as a fundamentally novel aspect of their approach. However, a recent publication in Nature Biotechnology, which they cite as a bioRxiv preprint, and whose computational tool (PEGG) they used to generate their pegRNA libraries, performed this exact approach (PMID: 38472508).

We address this point below.

(3) The ouabain co-selection approach is quite powerful but it has already been shown before to boost the selection of cells with functional prime editing activity (PMID: 36207338).

We included the indicated reference when introducing the co-selection strategy. We cannot find any examples of this strategy being used specifically in screening of variant effects, but as our work is indeed an extension of the co-selection strategy demonstrated, we have taken care to not make claims of broader novelty in the revised manuscript.

(4) While technically impressive, their screening data does not seem to provide additional biological insights beyond those already offered by variant effect prediction tools.

We appreciate the point the reviewer is making but respectfully disagree with this assessment. Having experimental data for many variants undeniably helps to disambiguate their functional effects. Experimental results are inherently orthogonal to variant effect prediction tools and thus can add independent value to interpretation. Experimental data are particularly valuable for variants with ambiguous effect predictions. To better illustrate this, we have functionally validated specific variants which are scored intermediately by computational predictors, as we detail below.

I believe points (1) and (4) above are important and could be boosted by the suggestions stated in (1) and below. Points (2) and (3) have to do with the novelty of the approach itself in the context of the current literature, including not referencing a published contemporary peer-reviewed study that used the same "ST" strategy described in point (2).

Major Comments:

(1) The authors describe in the discussion (Lines 451-460):

"In the context of well-established MAVEs for assessing variants by genome editing, such as base editing screens, saturation genome editing via HDR, and saturation prime editing, our platform offers the advantage of being able to install any set of short variants virtually anywhere in the genome, provided an active pegRNA can be designed.

Current limitations stem from the fact that only a limited fraction of pegRNAs lead to robust editing. This explains the incomplete scoring of variants for which we designed pegRNAs. As we show for variants in both SMARCB1 and MLH1, separation of signal from noise depends strongly on discerning more active pegRNAs, as editing percentages vary highly. In our platform, this is made possible via inclusion of STs with each pegRNA assayed."

The framing of the use of ST sites and the subsequent filtration of low efficiency pegRNAs as a fundamentally novel technological advancement seems inaccurate and should be revised. Specifically, a recent paper, whose bioRxiv preprint is cited by the authors and whose computational tool was used to generate the pegRNA libraries used in this study, described this exact methodology (PMID: 38472508). Given this contemporary study was peer-reviewed and published before Herger et al submitted their study to Cell Genomics (and to bioRxiv in parallel), their narrative should be modified to reflect the previous utilization of this method in the context of pooled prime editing. It seems dishonest to ignore this.

Thanks for this feedback. We are, of course, happy to include a reference to PMID: 38472508. When first introducing our ST strategy in **Results**, we did reference previous work from Sánchez-Rivera and colleagues describing the use of STs in base editing screens:

"We reasoned that an integrated readout of pegRNA activity may be important for accurate variant scoring due to the large variability of PE efficiencies within pegRNA pools. Therefore, downstream of each pegRNA, we included 55 bp of genomic target sequence to serve as a surrogate target (ST), capable of informing pegRNA editing efficiency after genomic integration (**Figure 1C**). STs have been previously employed for pooled base editing¹ and (pe)gRNA activity screens²."

We viewed this earlier base editing study as the key demonstration of STs for variant screening, though the use of STs for PE screening should clearly be included, too. This omission was simply from a failure to update references in the final days before submission. Considering our framing of STs as a concept already demonstrated by others, we do not feel we initially claimed this approach to be "fundamentally novel", though we now go further to make this more clear. Our goal in highlighting the STs in **Discussion** was to emphasize their importance to our HAP1 PE screening platform, including for both positive and negative selection screens, as this is essential info for others designing similar experiments.

We apologize for the outdated reference and have revised accordingly. Firstly, we have edited the penultimate paragraph of the introduction to reference more PE screening papers that have come out in recent months:

Until recently, it remained unclear whether these improved PE systems would enable scalable functional characterization of variants genome-wide. Saturation Prime Editing (SPE) was the first implementation of PE for assaying variant libraries²⁶. This approach was used to accurately identify pathogenic variants of interest but required testing individual pegRNAs for activity prior to library design and direct sequencing of edited loci for effect quantification. More recent studies have demonstrated that PE screening using lentiviral delivery of pegRNAs offers the promise of introducing virtually any short variant at any locus in a cost-effective manner^{27–33}. While this approach may prove ideal for testing large libraries of variants, most work in this area has been restricted to positive selection screening or limited to specific edit types to mitigate the challenge of inefficient PE. Indeed, to what extent PE screens can accurately distinguish human variants of clinical relevance remains an open question.

We have also added another reference to Gould *et al.* (PMID: 38472508) upon the first description of STs in **Results**:

STs have been previously employed for pooled base editing³⁸ and (pe)gRNA activity screens^{39,40}, and were recently shown to have high value in the context of pooled PE screening of *TP53* variants²⁸.

Lastly, we have revised our **Discussion** to make clear the value of STs for PE screening has been demonstrated elsewhere:

Current limitations stem from the fact that only a limited fraction of pegRNAs lead to robust editing. This explains the incomplete scoring of variants for which we designed pegRNAs. As we show for variants in both *SMARCB1* and *MLH1*, separation of signal from noise depends strongly on discerning more active pegRNAs, as editing percentages vary highly. In our platform, this is made possible via inclusion of STs with each pegRNA assayed. The improvement in data quality observed upon filtering pegRNAs by ST editing reflects the fact that ST editing is a reasonable proxy for ET editing, as we have shown (**Figures 2G and S5A**). Notably, the same approach of using STs (i.e., “sensors”) to filter inactive pegRNAs from analysis was also shown to improve PE screening results in a recent study of *TP53* variants²⁸.

(2) In the three screens that were performed (*SMARCB1*, *MLH1* coding, *MLH1* noncoding) there is no validation of individual pegRNAs. The authors primarily make use of variant effect predictors (CADD score, ClinVar annotation, SpliceAI) to "validate" the results of their screen. Validation of individual pegRNAs, including top hits that are not predicted to have a strong effect based on computational tools, would significantly add to this manuscript, and potentially provide novel biological insights. It is thus important that the authors perform experiments with individual pegRNAs and analyze the biological consequences of these perturbations, ideally in HAP1 cells but also in at least one diploid mammalian counterpart to illustrate that their methods are not strictly limited to haploid mammalian cells. This is important because readers may think that this technology is only useful at scale if one has to edit a single allele of a gene. However, a number of studies have shown that this is not the case for a handful of loci. In this regard, the amplicon sequencing of endogenous loci that is performed by the authors is insufficient to validate that the effects they show validate in a non-pooled format and produce biologically meaningful results.

We agree that showing more robust validation of individual variants, including across cell lines, would improve the paper considerably. We now present multiple experiments validating hits across both *SMARCB1* and *MLH1* in multiple ways. In summary, we have:

1. Tested a minipool of pegRNAs encoding depleted *SMARCB1* variants and validated the correct installment of edits and selection against LoF alleles. We also demonstrate that a newly identified intronic LoF variant leads to aberrant splicing by testing it individually.
2. Tested a minipool of newly designed pegRNAs encoding neutral and LoF *MLH1* variants and validated selection for LoF alleles upon 6TG treatment. Importantly, we also tested the same pool in K562 cells and show effects to be highly concordant between HAP1 and K562.
3. Demonstrated the molecular effects of two clinically relevant *MLH1* variants deemed LoF by introducing them individually to cells and observing aberrant splicing.

These results broadly confirm the assay's utility as a tool for discovery, yielding functional insights not clear from computational predictors alone while also shining light on some potential sources of noise in the screens, as to be expected in light of the imperfect separation of control variants. We describe these extensive new experiments in full below (please see section "**Detailed description of validation experiments**" on p.27), and

have added 2 supplementary figures, several paragraphs of **Results**, and complete descriptions of the work in **Methods**.

(3) The Github link provided is not publicly accessible and thus cannot be reviewed.

We apologize for this oversight. This GitHub page is now publicly accessible and updated through revision:

<https://github.com/FrancisCrickInstitute/PooledPEScreen>

(4) Supplementary Figure 8 shows that there are differences in the pegRNA function score between the PEmax and PEmax + MLH1dn conditions. The authors fail to address these differences, which are particularly relevant given that MLH1 is the gene being assessed, and presumably the MLH1dn protein could functionally and/or directly interact with MLH1 and modify any variant functions.

The reviewer raises a good point. The choice of whether or not to use the PEmax+MLH1dn HAP1 line for *MLH1* screening was something we thought about extensively but failed to address adequately before. Now, we more thoroughly detail our choice to perform screens in both HAP1:PEmax and HAP1:PEmax+MLH1dn, the difference in screen performance across the two lines, and our decision to combine data from experiments performed in both lines to produce final scores (while also making scores from individual experiments available).

We reasoned that expression of MLH1dn would increase editing rates, but we shared the concern that this might compromise the 6TG read-out. This is why we first asked whether stable MLH1dn expression in HAP1 confers the same degree of 6TG resistance as *MLH1* LoF mutations. The results of our 6TG titration (now **Figure S7A**) indicate MLH1dn expression does not confer the same degree of 6TG resistance as knockout of *MLH1*, but only partial resistance:

Figure S7. Effects of 6TG on growth and editing rates compared across HAP1 lines with modified MMR function.

(A) Viable cell counts are plotted by 6TG dose following 6 days treatment for HAP1:PEmax, HAP1:PEmax+MLH1dn, and HAP1:PE2+MLH1KO.

This led us to reason that LoF *MLH1* mutations introduced in HAP1:PEmax+MLH1dn would be enriched over neutral variants in our screens, as was observed. However, not knowing which line would perform optimally for screening, we still performed the *MLH1* screens in both HAP1:PEmax and HAP1:PEmax+MLH1dn for comparison, using a slightly higher dose of 6TG to perform selection in HAP1:PEmax+MLH1dn (detailed in **Methods**).

Our analyses indicate that the HAP1:PEmax line ended up performing best overall, as shown with the AUC analysis in **Figure 3B**. Yet, LoF variants were also clearly enriched in the HAP1:PEmax+MLH1dn line. One reason that the HAP1:PEmax+MLH1dn line still performed reasonably well was that editing rates were higher, as shown in **Figure S7B**. Given the reasonable function score correlations between cell lines (**Figures S9C and S9F**) and the expectation that there'll be some noise in each screen, we decided to produce final scores by averaging across lines (see **Methods**). Importantly, scores obtained in each individual line (HAP1:PEmax and HAP1:PEmax+MLH1dn) are also made available in **Tables S6 and S8**.

We have also now performed a more careful analysis of the data derived in HAP1:PEmax+MLH1dn to ask if it adds predictive power to the overall score set. Indeed, we observe a higher correlation to orthologous metrics of variant effect such as CADD and SpliceAI scores when data from both cell lines are averaged to produce final scores

(compared to using data from either line alone). We now include this analysis in **Figures S10A-C**. The improvement in correlation when averaging scores across lines makes sense considering these proof-of-concept screens display some noise, such that pooling data from more independent experiments is bound to improve accuracy.

The reviewer is right to point out that certain variants score differently between the two lines, but this is only a small subset of variants assayed. To enable more transparent comparison of scores between HAP1:PEmax and HAP1:PEmax+MLH1dn, we have added a heatmap showing scores of loss-of-function variants across the four experimental conditions (**Figures S10D and S10E**). Overall, big differences between cell lines are not common. As MLH1dn expression alone leads to lower 6TG sensitivity, we cannot exclude the possibility that variants scoring as LoF only when MLH1dn is expressed partially impair function.

To confirm whether the observed differences between lines were likely to be real, we chose 2 missense variants that scored highly in HAP1:PEmax+MLH1dn but not HAP1:PEmax to retest with new pegRNA designs in PE7-expressing cells without MLH1dn expression (**Figures S14C and S14D**). These variants were once more not enriched in cells lacking MLH1dn expression, suggesting expression of MLH1dn may contribute to the differences between lines.

On the whole, these new data indicate MLH1dn expression may impact screening results for a small subset of variants. As our focus is on establishing proof-of-concept for a new variant screening platform, we feel further investigation of the molecular basis of differences between cell lines with and without MLH1dn expression is beyond scope. Thinking forward, variants to be assayed in genes outside the MMR pathway are unlikely to be impacted by MLH1dn expression.

Figure S10. Consistency of MLH1 variant effects between HAP1:PEmax and HAP1:PEmax+MLH1dn.

(A-C) Correlation between function scores and CADD scores (top), and between function scores and SpliceAI scores (bottom) using data generated in HAP1:PEmax only (A), in HAP1:PEmax+MLH1dn only (B), or by combining data from both HAP1:PEmax and HAP1:PEmax+MLH1dn (C). Pearson correlation coefficients (r) are shown for each comparison. (D-E) Heatmaps of function scores across the four experimental conditions for all variants deemed LoF in the exon 10 screen (D) and for all variants deemed LoF in the non-coding screen (E) are shown. The color scale corresponds to the minimum and maximum score observed in any condition.

To make our scoring choice more transparent, we've modified our Results to explicitly report that optimal scoring is achieved when data from HAP1:PEmax and HAP1:PEmax+MLH1dn are combined in analysis, and that only a small subset of variants scored discordantly across lines.

Considering the possibility that MLH1dn expression may impact the effects of *MLH1* variants, we also computed cell-line specific function scores using data from only HAP1:PEmax or only HAP1:PEmax+MLH1dn. However, correlations to variant effect predictors were strongest when data from both cell lines were combined (**Figures S10A-C**). Though rare, a small subset of LoF variants displayed stronger effects with MLH1dn expression (**Figures S10D and S10E**), for example, c.866A>G and c.871T>A.

Additionally, we have modified the **Discussion** to summarize the trade-off between achieving optimal editing rates and producing the highest quality scores for variants in *MLH1*:

Whereas our data suggest that a small subset of *MLH1* variants may be influenced by expression of MLH1dn, for assaying variants in genes outside the MMR pathway, MLH1dn expression or *MLH1* knockout will likely increase editing without confounding interpretation.

Minor Comments:

In the first part of the manuscript, the authors conclude that MNVs are less efficiently installed than SNVs. However, the analysis provided in Figure 1H shows that a subset of MNVs are engineered much more efficiently than SNVs. Are there sequence-specific effects at play here? Please discuss.

This is a good question which we have now studied in more detail. In 27.9% (89/319) of comparisons, MNVs are installed more efficiently than corresponding SNVs. However, using an FDR threshold of 0.05 to define MNVs installed significantly more efficiently than SNVs yields only $n = 5$ MNVs, leaving us underpowered to extract pegRNA features that predict a large advantage of MNV installation.

Therefore, we asked whether relative MNV editing efficiency depends on either the total number of substitutions in the MNV or the maximum distance between substitutions. As shown in **Figure R2R.1**, no clear trends are apparent. 27.1% (70/258) and 31.1% (19/61) of pegRNAs installing two or three substitutions, respectively, achieve higher editing efficiency than the corresponding pegRNAs installing SNVs. Additionally, no clear trend is present when MNVs are grouped by the maximum distance between substitutions, except that MNVs are installed less efficiently when the maximum distance between variants is relatively large (7 or 8 nt). Top-performing MNV-pegRNAs are distributed evenly across these parameters.

Figure R2R.1. Comparison of SNV and MNV editing activity by pegRNA features. Log_2 -transformed ratios of correct day 10 ST editing for MNVs over corresponding SNVs, by total number of SNVs in the desired edit (A) and maximum distance between SNVs (B) in each MNV. (Boxplots: bold line, median; boxes, IQR; whiskers extend to points within 1.5x IQR.)

In conclusion, although some SNVs were more efficiently installed with co-edits than on their own, it is not clear from our data which sequence-specific features may be causing this. It is certainly possible that oligo design considerations for our libraries, such as limits on RTT length, are precluding the testing of more efficient MNV-pegRNAs. However, it is worth noting that inclusion of even synonymous co-edits can complicate functional variant interpretation, e.g. via unforeseen splicing alterations. Therefore, for the purpose of our screening platform, designing pegRNAs to install SNVs on their own offered both more efficient installation as well as more straightforward interpretation of variant effects. (As no clear trends emerged from our additional analyses, we've elected to not include these new plots in the revised manuscript.)

As mentioned above, have the authors performed any experiments in non-haploid cell types? How does the efficiency of the ouabain co-selection method compare if so?

We have not optimized ouabain co-selection in diploid lines, as our main focus has been extensive optimization of the platform in HAP1. In revision experiments, however, we did perform the same ouabain co-selection in K562, which produced successfully edited cells at a relatively high rate (**Figure S14A**). As our focus in this experiment was on validating the effects of individual edits across cell lines, we did not also maintain cells without co-selection to perform a comparison of editing rates. Nonetheless, this is evidence that the same co-selection strategy can be applied to non-haploid lines without causing, for instance, excess cell death.

Furthermore, as the original PE co-selection strategy was demonstrated in both K562 and HeLa³, we anticipate it should be broadly applicable to PE screening in other lines. However, as we haven't done a rigorous assessment of the co-selection strategy in lines beyond HAP1, we've refrained from commenting on the broader utility of this approach.

A comparison of the features that are predictive of correct editing (Figure 2E) to other works in the space (PRIDICT, DeepPrime) would be useful to understand whether these results are biased by the pegRNA design methods employed by the authors.

Given that these features do not appear to match up with these previous papers, a further explanation or exploration would be useful here both for the reader but also to lend more credential to the current study.

We agree with the reviewer that highlighting similarities and differences between our results and others would be useful. In summary, we observed a positive correlation between RTT homology length and pegRNA activity, as well as between PRIDICT score and pegRNA activity. Conversely, RTT length and nick-to-edit distance were both anti-correlated with pegRNA activity. Weak correlations were found for other features, including CRISPick score, PBS length and PBS GC content.

A careful comparison to other work reveals our results are generally confirmatory⁴⁻⁶. However, two main differences are worth explaining:

- 1.) PBS GC content has generally been found to be more strongly correlated with pegRNA efficiency than we observed. This difference is likely explainable by the high overall GC content of our *SMARCB1* target regions, 63.8% in total. This resulted in low sampling of PBS GC content below 50% (**Figure S4B**). Indeed, expanding our analysis to the *MLH1* pegRNA libraries we observe a stronger correlation between PBS GC content and PE efficiency (**Figures S4H and S4I**). Other key findings from our initial analysis of the larger *SMARCB1* dataset were replicated in this analysis, as well.

- 2.) We reported RTT length to be negatively correlated with pegRNA activity, while other studies⁴⁻⁶ have reported neutral or positive correlations. For RTT length specifically, it has been shown that a medium length is optimal, with both very short and long RTTs generally performing worse. However, the interplay between RTT length, RTT homology length and nick-to-edit distance can be complex, and interpretation is complicated by practical pegRNA design constraints, such as synthetic oligo size. To program edits far away from available PAMs, a greater nick-to-edit distance is required, demanding

extension of the RTT to allow for adequate RTT homology. Therefore, we expect that the negative correlation we observed between RTT length and pegRNA activity is a consequence of constraining RTT length to a maximum of 25 nts due to oligo size considerations (i.e., synthesis of oligo pools). This is demonstrated in **Figure R2R.2** below, which shows that pegRNAs nicking far from the edit generally had shorter RTT homologies and consequently achieved very low editing efficiencies. In attempting to perform saturation mutagenesis, many variants far from suitable PAM sites inherently require longer RTTs, but these perform poorly compared to variants near suitable PAM sites for which RTT homology length was typically greater.

Fig. R2R.2. Relationship of nick-to-edit distance, RTT homology length and pegRNA editing efficiencies in the *SMARCB1* pegRNA library.

(A) A heatmap of correct ST editing rates on D10 in relation to nick-to-edit distance and RTT homology length for all designed pegRNAs. (B) A heatmap showing the total number of pegRNAs for which ST editing was measured.

In summary, as the reviewer alludes to, our results are dependent on the more confined pegRNA feature space explored in our libraries. To address this point, we have added a supplemental figure to offer more transparency on the pegRNA parameters sampled in our library (**Figure S4**) and revised the following paragraph in **Results**:

We next asked which pegRNA features correlated with higher PE efficiencies. Of pegRNA features analyzed, RTT homology length (distance from edit to 5' end of RTT) was most strongly correlated with ST editing on day 10 ($R = 0.43$, $r = 0.37$), followed by PBS length and PBS GC-content (**Figures 2E and S4A-G**). On the contrary, total RTT length and nick-to-edit distance were negatively correlated with ST editing. We also observed a modest correlation with CRISPick scores (i.e., predictions of gRNA on-target activity)^{45,51}. We retrospectively computed PRIDICT scores³⁶ where possible for our pegRNAs and found a reasonable correlation with experimentally determined activities, demonstrating

the value of this tool for future pegRNA design. Notably, the features of pegRNAs most predictive of editing activity were highly consistent across libraries used in this study (Figures S4H and S4I). Overall, these results are generally confirmatory of prior findings^{28,36,37}. Minor differences, such as the inverse correlation between RTT length and editing efficiency, likely stem from pegRNA design limitations imposed by capping synthetic oligo length at 243 nt.

Figure S4. Impact of pegRNA features on correct ST editing.

(A-G) Percentage correct ST edits at day 10 averaged across the two non-ouabain treated experiments for $n = 8,612$ *SMARCB1*-targeting pegRNAs stratified by PBS length (A), PBS GC content (B), RTT length (C), (distal) RTT homology length (D), nick-to-edit distance (E), CRISPick score (F), and PRIDICT score (G). Points are colored by Gaussian kernel density estimation and a linear regression model (red line) with 95% confidence interval band (gray) is shown.

(H and I) Spearman correlations of percentage of correct ST edits at day 10 with different pegRNA features for $n = 2,695$ and $n = 3,747$ pegRNAs in the *MLH1* exon10 (H) and *MLH1* non-coding libraries (I), respectively.

Detailed description of validation experiments

To experimentally validate effects of select variants assayed in our large-scale screens, we performed a series of experiments to re-test cellular and molecular effects of variants in both *SMARCB1* and *MLH1*.

SMARCB1

First, for *SMARCB1* we selected 9 variants to re-test. This included 1 neutral synonymous variant (p.Thr335= / "V1") and 2 expected-LoF variants (c.1119-1G>C / "V2", a canonical splice variant and c.Pro334* / "V3") as controls. The remaining 6 variants scored as LoF in the original screen, including 3 missense, 1 intronic, and 2 3'UTR variants. The pegRNAs that most efficiently installed each variant in the large screen were individually cloned and used for lentivirus production. We then tested this pegRNA set as a small pool (i.e. a "minipool") using an independently derived HAP1 line in which *MLH1* was knocked out and PE7⁷ was expressed instead of PEmax.

Analysis of ST editing 4 days post-transduction revealed all but one pegRNA (c.*71A>C) achieved higher editing efficiency in HAP1:PE7+MLH1KO compared to HAP1:PEmax+MLH1dn (**Figure S6A**). The cell pool was split in two on day 4 post-transduction to treat one population with ouabain until day 10 for co-selection. Cells were cultured until a final harvest on day 32, followed by NGS preparation, sequencing, and function score calculation, as before. With relatively few variants in the pool, we were able to accurately quantify variants' endogenous frequencies to directly assess depletion of LoF variants.

We observed strong correlations for both pegRNA-derived and ET-derived function scores across samples with and without co-selection (Pearson's $r = 0.97$ and $r = 0.99$, respectively; **Figures S6B and S6C**). Furthermore, function scores derived from pegRNA frequencies and ET variant frequencies were also highly correlated (Pearson's $r = 0.99$; **Figure S6D**), thus demonstrating depletion of active pegRNAs closely correlates with depletion of installed LoF variants.

The minipool data confirmed the effects of most but not all LoF variants in the pool, including c.1119-1G>C, p.Pro334*, p.Gln368His, p.Asp369Ala, and c.1119-12C>G. Importantly, the disruptive effect of c.1119-12C>G, a ClinVar-VUS located in intron 8 was confirmed, thus providing functional evidence supporting this variant's pathogenicity.

Determining effects of variants individually, as the reviewer suggested, is challenging for variants leading to cell death. We attempted this by introducing the same pegRNAs

individually to HAP1:MLH1KO cells lacking constitutive prime editor expression. After transduction, we transfected cells with pCMV_PE7-BSD and applied Blasticidin selection for 5 days to induce a window of editing. By transiently delivering PE7, we aimed to prevent the continuous installation of edits (which confounds measurement of variant depletion in negative selection experiments). Using this approach, we found neutral variant frequencies remained essentially unchanged from days 10 to 32 post-transfection (**Figure S6E**). By day 10, variants were installed endogenously at percentages ranging from 12.8 to 70.9%. We again calculated function scores for variants, this time normalizing to the fraction of unedited alleles. These scores correlated highly with function scores from the minipool experiment, once more validating LoF effects of the same 5 variants (Pearson's $r = 0.91$; **Figure S6F**).

Given the clinical relevance of the intronic VUS confirmed to be LoF, c.1119-12C>G, we next asked whether aberrantly spliced transcripts were detectable in the population of cells to which the pegRNA installing c.1119-12C>G was introduced. Indeed, amplicon sequencing of *SMARCB1* transcripts revealed a novel splice junction only present in cells receiving the variant (**Figure S6G**). This junction is located adjacent to the installed variant and leads to 11 bp of intron retention, thus corroborating the effect of this variant on the molecular level. We note that the fraction of transcript reads mapping to this junction is only 1.9%. This is to be expected given most cells sampled were unedited and aberrantly spliced transcripts will be subject to nonsense mediated decay. The combination of this new splicing event being observed at the exact location of the variant and only in edited cells plus the reduced cellular fitness caused by the variant strongly corroborates its LoF effect.

Among the 3 variants deemed LoF in the original screen that did not validate, p.T372S showed a mildly negative effect on growth across validation experiments. While this was not strong enough to confirm the variant as LoF, it is possible that V6 exerts an intermediate effect on *SMARCB1* function. Indeed, the conservative p.T372S substitution yields a lower AlphaMissense score (0.987) than the two other missense variants re-tested, p.Gln368His (0.996) and p.Asp369Ala (1.000), which were both confirmed to be LoF.

The other two variants that scored as LoF previously but did not validate were c.*65T>G and c.*71A>C. As a class, 3'UTR variants are generally under relatively little selective constraint⁸, consistent with only 2 of 49 *SMARCB1* 3'UTR variants scoring as LoF in our screen. These 2 variants also have much lower CADD scores than other variants scored LoF (15.0 and 12.4, respectively), suggesting they may be false positives.

Notably, the nearby 3'UTR variant, c.*82C>T, is deemed “likely pathogenic” in ClinVar and associated with schwannomatosis, though its causal mechanism has yet to be defined⁹. This motivated our decision to re-test V8 and V9 because validating potential LoF effects could potentially reveal new mechanisms underlying the role of 3'UTR variants in disease. Ultimately, however, neither V8 nor V9 were depleted in validation experiments, confirming these to be likely false positives.

Taken as a whole, these experiments confirm our *SMARCB1* screen identified new variants leading to loss-of-function with reasonable accuracy for a proof-of-concept study, while also illustrating the importance of careful validation, particularly for variants with relatively low prior probabilities of being functionally impactful.

Figure S6. *SMARCB1* LoF variant validation.

(A) Comparison of ST editing efficiencies previously determined in HAP1:PEmax+MLH1dn and newly generated HAP1:PE7+MLH1KO cell lines 4 days after transduction with a pool of 9 pegRNAs used for variant effect validation. (Boxplot: bold line, median; boxes, IQR; whiskers to points within 1.5x IQR.) (B and C) Function scores for the minipool experiment were calculated as log₂-ratios of pegRNA frequencies

(B) or ET variant frequencies (C) at day 32 over day 10, normalized to the score of a synonymous variant (p.Thr335= / “V1”) scored neutrally in the screen. Function scores are strongly correlated between conditions with (+obn) and without co-selection.

(D) Function scores derived from pegRNA frequencies and ET variant frequencies (averaged across conditions) are strongly correlated.

(E) Fraction of aligned sequencing reads corresponding to correct edits, unedited reference sequence (“wildtype”), and other editing outcomes as determined by CRISPResso2 analysis for samples to which individual pegRNAs were introduced. Data are shown for 3 timepoints.

(F) Endogenous function scores determined by testing pegRNAs individually (i.e., “arrayed”) are well-correlated to endogenous function scores from minipool screening (i.e., “pooled”). (G) A potential splicing defect caused by c.1119-12C>G (“V7”) was evaluated with RT-qPCR, gel electrophoresis and NGS using cells to which either the V1 (control) or V7 pegRNA was introduced individually. Bands corresponding to RT-qPCR products across exon junctions 6-7, 8-9 and 7-9 are shown on the gel. An additional primer set was used to selectively amplify an 11-bp product of intron retention (“IntRet”) predicted to be caused by V7. Red arrows indicate observed differences between V1 and V7 consistent with aberrant splicing caused by the latter. NGS analysis of the ex7-9 products confirmed the presence of the IntRet isoform at 1.9% for V7 compared to 0.0% for V1 (averaged across $n = 2$ technical replicates).

These findings have been integrated to **Results** as follows:

Among 164 variants scored with this filter, we identified 12 significantly depleted variants, including 3 nonsense and 2 canonical splice variants, using a false discovery rate (FDR) of 0.05 to call LoF variants (see **Methods**). LoF missense variants clustered within a highly conserved α -helix of the SMARCB1 CTD. We also scored the intronic SNV c.1119-12C>G as LoF, which is annotated as a VUS in ClinVar⁷ but has been hypothesized to disrupt splicing⁵² and has a SpliceAI score⁴ of 0.96.

To validate these findings, we selected 8 significantly depleted variants for further analysis, including 3 missense variants, the intronic variant c.1119-12C>G, and 2 3'UTR variants. These variants were re-assessed in a newly generated PE7-expressing HAP1 line with *MLH1* knocked out (HAP1:PE7+MLH1KO). Lentiviruses encoding pegRNAs programming each variant were mixed and assayed in a small pool (i.e., a “minipool”) following the same procedure as for the large-scale screen. With this approach, we observed modestly higher ST editing rates for the same set of pegRNAs compared to before (day 4 median 78.2% versus 64.9% previously; **Figure S6A**). Function scores derived from pegRNA frequencies and endogenous variant frequencies with and without co-selection were highly correlated (Pearson's $r = 0.97$ to 0.99 ; **Figures S6B-D**), and LoF effects of most but not all variants previously deemed LoF were confirmed, including one nonsense variant, one canonical splice variant, 2 of 3 missense variants, and the intronic VUS located in intron 8. Despite the reported pathogenicity of a nearby 3'UTR variant in predisposition to schwannomatosis⁵³, we did not observe LoF effects for the 2 3'UTR variants re-tested, indicating they were likely false positives.

Additionally, we assayed the same variants individually by transducing HAP1:MLH1KO cells with each pegRNA and then transiently expressing PE7. This approach confirmed the LoF variants selected against in the minipool to be steadily depleted, in contrast to neutral variants (**Figure S6E**). Function scores determined by individually assaying variants correlated highly with function scores from the minipool experiment (Pearson's $r = 0.91$; **Figure S6F**), further validating the LoF effects. Lastly, given the clinical relevance of the intronic VUS, c.1119-12C>G, we assessed this variant's impact on splicing. Amplicon sequencing of *SMARCB1* transcripts revealed a novel splice junction unique to cells harboring c.1119-12C>G (**Figure S6G**). This new splice junction is immediately adjacent to c.1119-12C>G and leads to 11 bp of intron retention, corroborating this variant's effect at the molecular level.

Overall, these examples illustrate how select LoF variants efficiently installed by PE can be functionally identified via negative selection and suggest that further improvements to editing efficiency will enable many more variants to be assessed in this manner.

MLH1

To understand potential sources of error in the *MLH1* screens and to address reviewer comments regarding variant validation, we re-tested a small panel of *MLH1* variants using new pegRNA designs in both HAP1 and K562 cells expressing PE7⁷. It is worth noting that K562 has 4 copies of *MLH1* per DepMap¹⁰. Therefore, while we think it is valuable to use K562 to test a small number of variants, we do not anticipate our larger screening approach would work robustly in this line without extensive optimizations beyond the scope of our manuscript.

Our panel of variants to be re-tested included:

1. Neutral variants expected to not be enriched during 6TG selection: c.870A>G ("V1"; a synonymous variant) and c.116+220T>G ("V2"; a deep intronic variant classified as benign in ClinVar).
2. Variants scored as LoF in the initial screen predicted to induce 6TG resistance upon re-testing: c.829G>T / "V3", c.116+1G>A / "V4", c.-7_1del / "V5", c.884+3A>G / "V6", c.885-3C>G / "V7", and c.1732-264A>T / "V8". This group includes variants strongly predicted to be LoF, c.829G>T, a nonsense variant and c.116+1G>A, a canonical splice site variant, as well as a ClinVar-pathogenic variants we had previously highlighted, c.-7_1del and c.1732-264A>T. The group also includes noncoding variants of unknown function, such as splice regions variants c.885-3C>G (highlighted in the initial manuscript) and c.884+3A>G. At

the time of library design, c.884+3A>G had conflicting interpretations in ClinVar, but is now classified as pathogenic/likely pathogenic.

3. Variants upstream of *MLH1* coding sequence which we suspected may represent false positives due to having low CADD scores: c.-219G>T / “V9”, c.-213G>A / “V10”, and c.-3A>T / “V11” (CADD scores ranging from 7.4 to 20.7).
4. Two missense variants from exon 10 that were enriched in the screen performed in HAP1:PEmax+MLH1dn but not HAP1:PEmax: c.866A>G / “V12” and c.871T>A / “V13”.

pegRNAs for all 13 variants were cloned individually and combined into a minipool to allow for assessment of ST editing and pegRNA selection, as well as ET editing and ET variant selection. The timeline of this experiment closely followed that of the original screen, with ouabain co-selection from D13 to D18 and 6TG selection from D19 to D33.

The combination of using pegRNAs designed with PRIDICT2 and cells expressing PE7 led to a higher fraction of newly designed pegRNAs passing the 5% ST editing threshold than observed in the original screen, with only 1 pegRNA (c.1732-264A>T) showing low editing at 19 days post transduction (**Figure S14A**). This pegRNA was removed from analysis. Across biological replicates and across both HAP1 and K562, editing rates and pegRNA selection were highly correlated (**Figures S14A-D**).

The synonymous and intronic variants previously scored as neutral were not enriched during 6TG selection (function scores -0.28 to 0.28 and -0.14 to 0.14 in HAP1 and K562, respectively; **Figure S14C**), whereas the nonsense and canonical splice variants were (function scores 5.65 to 5.65 and 2.64 to 3.06, in HAP1 and K562, respectively). The same selection patterns were also confirmed by sequencing ETs (**Figure S14D**). The two splice region variants that had scored highly (c.884+3A>G and c.885-3C>G) were once again strongly enriched at both pegRNA and ET levels, whereas the newly designed pegRNA installing the start-loss variant (c.-7_1del) was only mildly enriched (**Fig. S14C**). Notably, this pegRNA produced less efficient editing (**Fig. S14A**), potentially explaining its more mild enrichment compared to pegRNAs installing other LoF variants. Indeed, ET sequencing revealed comparable selection for V5 compared to the other LoF variants included in the minipool (**Figure S14D**). Overall, these results confirm that these 5 variants originally scored as LoF all lead to 6TG resistance when introduced with orthologous pegRNAs.

Notably, the two pegRNAs encoding missense variants (c.866A>G and c.871T>A) were once more not enriched upon 6TG selection. This result is consistent with the original screen result from HAP1 cells lacking MLH1dn and suggests that a small subset of variants' functional consequences may be influenced by expression of MLH1dn (as discussed more in response to the reviewers' questions on MLH1dn expression).

Meanwhile, the 3 pegRNAs that introduced variants upstream of the CDS (c.-219G>T, c.-213G>A, and c.-3A>T) were not enriched upon 6TG selection in this experiment, suggesting these variants were likely false positives in the original screen. While we do not observe a large number of false positives overall (as evidenced by very few ClinVar-benign variants scoring highly), we speculate that the original pegRNAs used to install these variants may have led to unintended effects, such as CRISPR inhibition or the creation of indels. These cautionary examples indicate that validating variant effects with orthologous pegRNAs can play an important role in reducing false positives. Therefore, we have revised a paragraph of the **Discussion** to specifically caveat both false positives and false negatives and the importance of validating findings:

Our work also illustrates the importance of validating functional effects of pegRNAs to confirm variant effects, as both false positives and false negatives can occur in pegRNA screening. We used multiple approaches to validate variant effects, including sequencing endogenous targets, replicating results with orthologous pegRNAs tested in small batches or individually, and confirming effects across multiple cell lines. Going forward, we expect improved pegRNA design will likely allow multiple, orthogonally designed pegRNAs per variant to be tested, either together or in successive experiments. With more data, there will also be opportunities to refine function scores to better reflect variant-level confidence, incorporating both strength of pegRNA selection and ST editing rate. For assays of variants observed in patients, there will remain a need for rigorous clinical benchmarking prior to using assay results to aid variant classification.

As further validation, we aimed to investigate the specific molecular effects of two LoF variants on splicing. The first variant, c.884+3A>G, is located 3 bp downstream of exon 10, while the second, c.885-3C>G, is located 3 bp upstream of exon 11. Both variants have intermediate SpliceAI scores, 0.29 and 0.73, respectively, and had uncertain or conflicting pathogenicity annotations at the time of initial screening. For molecular validation of effects, we transduced HAP1:PE7 cells with individual pegRNAs and enriched successfully edited cells via 6TG selection. Sequencing of amplicons spanning exons 8 to 12 revealed that both variants cause aberrant splicing (**Figures S14E-G**). Specifically, c.884+3A>G (V6) results in near-complete loss of the reference transcript (3.3% and 2.8% in replicate experiments compared to 65.6% and 62.1% in controls).

Multiple alternative splicing events are seen at higher levels, including skipping of both exons 9 and 10 or exon 10 alone, with or without utilization of an alternative splice acceptor site 5 bp from the canonical exon boundary. Notably, c.884+3A>G (V6) has recently been linked to exon 10 skipping¹¹, and is now reclassified as pathogenic/likely pathogenic in ClinVar, consistent with our findings. Similarly, c.885-3C>G (V7) leads to a marked reduction in canonical transcript (11.7% and 38.0% in two replicates). The three most common alternative transcripts include skipping of exon 11, combined skipping of exons 9 and 11, and combined skipping of exons 10 and 11. It is worth noting that replicate 2 exhibited a lower fraction of correctly edited cells (88.9% in replicate 1 vs 76.0% in replicate 2), consistent with the higher fraction of canonical transcripts observed in replicate 2.

These findings are incorporated to the manuscript at the end of **Results** as follows:

To validate LoF effects of *MLH1* variants, we first performed amplicon sequencing of edited regions, using the same gDNA from HAP1:PEmax cells from which pegRNAs were sequenced. Indeed, LoF variants including c.885-3C>G, c.1732-264A>T, and c.-7_1del were strongly enriched in their respective regions post-6TG selection, corroborating pegRNA-based function scores (**Figures S12C and S12D**). The lack of enrichment seen for nearly all SNVs in the 5'UTR and upstream region was consistent with the low function scores of these variants. Notably, one variant in close proximity to the TSS, c.-219G>T, was deemed LoF in the screen but not strongly enriched in ET sequencing, indicating that its pegRNA-derived function score may be conflated by CRISPR inhibition (CRISPRi) or some other unintended effect.

We next selected 13 variants for further characterization using newly designed pegRNAs, including 2 neutral variants and 2 variants strongly expected to be LoF. Other variants re-tested in this group included the start-loss variant (c.-7_1del), 2 splice region variants (c.884+3A>G and c.885-3C>G), the deep intronic variant c.1732-264A>T, the upstream variant c.-219G>T, 2 variants in the 5'UTR (c.-213G>A and c.-3A>T), and 2 missense variants strongly enriched in HAP1:PEmax+MLH1dn but not in HAP1:PEmax (c.866A>G and c.871T>A). Lentivirus was produced for each pegRNA and combined to generate a minipool which was then assayed in both HAP1 and K562 cells expressing PE7.

Across biological replicates and across both HAP1 and K562, ST editing rates and pegRNA selection were highly correlated (**Figures S14A and S14B**). Several variants that previously scored as LoF were again enriched upon 6TG treatment, including both expected-LoF variants, the start-loss variant c.-7_1del, and both splice region variants (**Figures S14C and S14D**). Meanwhile, the upstream and 5'UTR variants were not enriched in these experiments using independent pegRNAs, consistent with their lack of enrichment at the endogenous locus (**Figure S12C**) and confirming they were likely false

positives in the larger screen. Notably, both missense variants re-tested were previously only enriched in HAP1 lines expressing MLH1dn (**Figures S10D and S10E**). In this experiment using HAP1 and K562 lines lacking MLH1dn, once more neither variant was enriched, suggesting MLH1dn expression may have impacted how these *MLH1* variants were scored.

Among variants confirmed to be LoF, c.884+3A>G and c.885-3C>G both have intermediate SpliceAI scores of 0.29 and 0.73, respectively. Given the clinical importance of these variants, we further investigated their molecular effects on splicing by individually transducing HAP1:PE7 cells with each corresponding pegRNA and enriching for successfully edited cells via 6TG selection. Gel electrophoresis and amplicon sequencing revealed that both variants drastically reduce canonical transcript isoforms and increase products of aberrant splicing (**Figures S14E-G**). Notably, c.884+3A>G has recently been linked to exon 10 skipping⁵⁵ and reclassified as likely pathogenic in ClinVar, consistent with our findings, whereas c.885-3C>G remains a VUS for which our functional data may prove valuable.

In summary, these experiments firmly establish that our PE platform can be used to identify LoF variants acting via diverse mechanisms across large non-coding regions. However, careful validation, for instance with ET sequencing or use of independent pegRNAs and cell lines, is critical for improving the accuracy of screening.

Figure S14. Experimental validation of *MLH1* variant effects.

(A) Correlations of correct ST-editing percentages on day 19 across replicates and cell lines for $n = 13$ variants re-tested in the *MLH1* minipool with newly designed pegRNAs. Pearson correlation coefficients (r) are shown.

(B) Correlations of function scores across replicates and cell lines are shown for $n = 12$ variants in the minipool with Pearson correlation coefficients (r) for each comparison. V8 was excluded from analysis due to low editing efficiency (less than 5% ST editing on D19).

(C) Final function scores were calculated by averaging function scores across HAP1:PE7 replicates (left) or K562:PE7 replicates (right), and by normalizing to the median of neutral controls (V1 and V2). (D) Endogenous function scores are shown for $n = 11$ variants tested in HAP1:PE7 (excluding V8 due to insufficient editing activity and V4 due to lack of suitable primers).

(E-G) Potential splicing defects caused by V6 (c.884+3A>G) and V7 (c.885-3C>G) were investigated using RT-qPCR, gel electrophoresis and NGS. RT-qPCR products for V2 (a neutral control) V6, and V7 were generated using primers binding within exon 8 and across the exon 11-12 junction (E). NGS analysis of the resulting products for V6 (F) and V7 (G) reveals the loss of reads corresponding to canonical splicing and increased exon-skipping events. The three most abundant non-canonical isoforms

detected are shown for each of V6 and V7, with the percentage of reads matching each product provided for each of $n = 2$ biological replicates compared to control (V2).

Our **Methods** have been revised in line with STAR Methods formatting. Additional sections have been added to describe all experiments performed in revision, with updated sections reproduced below:

Plasmids

For stable integration of PEmax with and without MLH1dn, a lentiviral vector was prepared by PCR amplification of the corresponding cassettes from pCMV-PEmax-P2A-hMLH1dn (#174828, *Addgene*) and subsequent insertion into pLenti-PE2-BSD (#161514, *Addgene*), which had been digested with EcoRI and XbaI (#R0145S & #R3101S, *NEB*). A lentiviral construct for stable integration of PE7 (pLenti-PE7-2A-BSD) was created by digestion of pLenti_PEmax-2A-BSD with BamHI and XbaI (#R0145S & #R3136S, *NEB*). PEmax was replaced with PE7, which was amplified from pT7-PE7 for IVT (#214813, *AddGene*), via Gibson assembly. For transient expression of PE7, pCMV_PE7-2A-BSD was constructed by Gibson assembly of a PCR-linearized pCMV-PEmax-P2A-hMLH1dn backbone with the PCR-amplified PE7-2A-BSD cassette from pLenti-PE7-2A-BSD. Single pegRNAs were stably integrated using lentiviral vectors cloned by assembling pegRNA oligo duplexes into BsmBI-digested (#R0739L, *NEB*) Lenti_gRNA-Puro (*Addgene* #84752). The dual-pegRNA lentiviral vector, used for pegRNA library cloning, was prepared by amplification of a gene fragment encoding the mU6 promoter and a BsmBI cloning site (*Twist Bioscience*) followed by assembly into a KfiI- and Eco72I- digested (#FD2164 & #FD0364, *Thermo Scientific*) Lenti_gRNA-Puro plasmid with a pre-inserted co-selection pegRNA under the hU6 promoter. For bacterial amplification of pegRNA scaffolds, oligo duplexes, encoding the different scaffold designs (original, F+E, F+E v1, F+E v2) and flanking BsmBI recognition sites, were inserted into the KpnI-digested (#R3142S, *NEB*) pU6_pegRNA-GG-Acceptor plasmid via Gibson assembly. All constructs were propagated in NEB Stable Competent *E. coli* (High Efficiency; #C3040H, *NEB*) unless stated otherwise.

Cell lines and culture

... K562 cells were cultured at 37 °C with 5% CO₂ in Roswell Park Memorial Institute (RPMI) medium (#11875093, *Gibco*) supplemented with 10% FBS and 1% Pen-Strep. Cell density was maintained below 0.8×10^6 cells/mL. For culture of PE-expressing HAP1 and K562 lines, media was additionally supplemented with 5 µg/mL or 10 µg/mL blasticidin S (R21001, *Gibco*) respectively. All cell lines were confirmed to be free of mycoplasma.

Generation of PE cell lines

... The HAP1:PE2+MLH1KO line was used only for titering 6TG dose. It was created by first transducing parental HAP1 cells with pLenti-PE2-BSD at low MOI (as for PEmax cell lines), then transfecting successfully transduced cells with a pX459 construct targeting *MLH1*. Individual clones were isolated and screened by Sanger sequencing, and a line

with a frameshifting indel in exon 18 was selected. The HAP1:PE7+MLH1KO cell line used for validation experiments was created by first transfecting parental HAP1 cells with a pX459 construct targeting *MLH1*, followed by sequencing to confirm a clonal population with a 1 bp insertion in *MLH1* exon 10. This clone, HAP1:MLH1KO, as well as parental HAP1 and K562 cells were then transduced with pLenti_PE7-2A-BSD. Spinfection of K562 cells was performed by centrifugation of virus with cells at 1,321 x g for 1.5h at 37 °C. Cells were selected for 10 to 14 days with 5 µg/mL (HAP1:PE7) or 10 µg/mL (K562:PE7) blasticidin S and subsequently used for variant effect validation experiments.

Calculation of pegRNA scores and function scores

... To compare function scores derived in HAP1:PEmax and HAP1:PEmax+MLH1dn, cell line-specific scores were calculated. First, condition-specific scores were calculated and normalized as described above. Then, the normalized scores were averaged across both conditions using the same cell line and normalized to the median score of synonymous variants (for the *MLH1* exon 10 screen) or to the median score of intronic variants (for the *MLH1* non-coding screen).

Variant effect validation experiments

Effects of select variants were reassessed individually or as part of a small pool (i.e., a “minipool”).

For *SMARCB1*, 9 variants in total were selected for experimental validation, including 1 expected neutral control (p.Thr335=), 2 expected LoF variants (c.1119-1G>C and p.Pro334*), and 6 variants across coding, intronic and 3'UTR regions (p.Gln368His, p.Asp369Ala, p.Thr372Ser, c.1119-12C>G, c.*65T>G, and c.*71A>C). For each variant, the pegRNA with the highest activity as defined by ST editing in the screen was used for the validation experiments.

For *MLH1*, 13 variants were selected, including 2 expected neutral controls (a synonymous variant, c.870G>A, and c.116+220T>G), 2 expected LoF variants (a nonsense variant, c.829G>T, and c.116+1G>A), and 9 variants across upstream, 5'UTR, coding and intronic regions (c.-7_1del, c.884+3A>G, c.885-3C>G, c.1732-264A>T, c.-219G>T, c.-213G>A, c.-3A>T, c.866A>G, and c.871T>A). For these *MLH1* variants, new pegRNAs were designed using PRIDICT2⁴⁰.

Selected pegRNA-ST cassettes were ordered and synthesized as custom gene fragments (*Twist Bioscience*). Cloning of pegRNAs into lentiviral vectors proceeded as before, except that gene fragments were directly used for Gibson assembly without PCR amplification and the pegRNA scaffold was included in synthesis. Lentivirus particles were produced

for individual pegRNA plasmids, as described above, and mixed at equal concentrations to generate the minipool for each gene.

SMARCB1 variant effect validation:

The procedure for testing the *SMARCB1* minipool remained largely unchanged from the large-scale experiments. In summary, 0.8×10^6 HAP1:PE7+MLH1KO were transduced at low MOI (0.1-0.3), followed by selection with puromycin for 3 days. On day 4 post-transduction, cells were split in half with 1 cell pool undergoing co-selection with 5 μ M ouabain for 6 days.

For assaying individual pegRNAs in an arrayed format, 0.8×10^6 HAP1:MLH1KO cells per pegRNA were transduced at low MOI, followed by puromycin selection for 3 days. 0.8×10^6 transduced cells were then transfected with pCMV_PE7-2A-BSD using Xfect Transfection Reagent, followed by selection with 5 μ g/mL blasticidin for 6 days.

Cell populations were maintained until day 32 following delivery of editing reagents, with splitting and sampling every 2-3 days. Amplification from gDNA and sequencing of endogenous targets and pegRNA-ST cassettes (for minipool experiments only) was performed as described above.

MLH1 variant effect validation in HAP1:PE7 cells:

Two replicates of 2×10^6 HAP1:PE7 cells were transduced with a pegRNA minipool at low MOI (0.1-0.3) and selected with puromycin for 3 days. Co-selection with 5 μ M ouabain was performed from day 13 to day 18 post-transduction. On day 19, functional selection with 1.2 μ M 6TG was initiated and continued until day 33. Cell pellets were harvested before and after 6TG selection.

To determine splicing consequences of c.884+3A>G and c.885-3C>G, 2×10^6 HAP1:PE7 cells were transduced with individual pegRNAs encoding a control variant (c.116+220T>G), c.884+3A>G, or c.885-3C>G, then selected with puromycin for 3 days. On day 7, each cell population was split in two and 6TG selection was performed until harvesting once cells reached confluency (day 22-30).

MLH1 variant effect validation in K562:PE7 cells:

Two replicates of 0.5×10^6 K562:PE7 cells were transduced with the *MLH1* pegRNA minipool at low MOI (0.1-0.3) using spinfection (centrifugation at $1,321 \times g$ for 1.5h at 37C). On day 2 post-transduction, the media was replaced and the cells were selected with 2 μ g/mL puromycin for 5 days. Co-selection with 5 μ M ouabain was performed from day 13 to day 18. On day 19, 6TG selection was initiated at 1.2 μ M and continued until day 33. Cells were harvested before and after selection. Dead cells were removed prior to cell harvest on day 33 using Dead Cell Removal Kit (#130-090-101, Miltenyi Biotec).

Detection of aberrant splice products:

The *SMARCB1* variant c.1119-12C>G and *MLH1* variants c.884+3A>G and c.885-3C>G were selected for mechanistic follow-up to investigate potential changes in splicing. These three variants were experimentally deemed LoF but reported in ClinVar to be of unknown or conflicting clinical significance when screens were initiated in November 2022. As of December 2024, *SMARCB1* c.1119-12C>G and *MLH1* c.885-3C>G remain VUS, whereas c.884+3A>G is now deemed likely pathogenic. To assess these variants' impacts on splicing, total RNA and gDNA were extracted from HAP1 samples to which individual pegRNAs were delivered using the AllPrep DNA/RNA Mini Kit (#80204, Qiagen) with QIASHredder columns (#79656, Qiagen).

For *SMARCB1*, 100 ng total RNA from day 10 samples corresponding to a neutral control variant (p.Thr335=) and c.1119-12C>G was subjected to RT-qPCR using Luna Universal One-Step RT-qPCR Kit (#E3005S, NEB) in technical duplicates with custom primer sets designed using the Primer3 webtool (**Table S9**). Amplicons obtained from limited-cycle PCR were analyzed by gel electrophoresis using an E-Gel system (#G401002, Invitrogen). Amplicons spanning exons 7 to 9 (Ex7-9 amplicons) were prepared for NGS and sequenced on an Illumina MiSeq. The most frequent sequencing reads were counted and mapped against the reference transcript sequence to identify distinct splicing outcomes between control and c.1119-12C>G samples. Frequencies of splice products were averaged across technical replicates.

For *MLH1*, 2 µg of total RNA extracted from samples corresponding to the post-6TG selection timepoints of the neutral control variant (c.116+220T>G), c.884+3A>G, and c.885-3C>G were reverse-transcribed using SuperScript IV First-Strand Synthesis System and oligo(dT) primers (#18091050, Thermo Fisher). Subsequently, the RT product was quantified using the Qubit dsDNA HS kit (#Q33231, Thermo Fisher) and 40 ng from each sample was used as input for PCR. Primers were designed to bind within exon 8 and across the exon junction 11-12. Amplicons were visualized via gel electrophoresis and sequenced on an Illumina MiSeq instrument. Paired-end sequencing reads were demultiplexed and merged, and the most frequent reads for each sample were determined and mapped against the reference transcript to quantify the proportion of each splice product.

Materials availability

Plasmids generated in this manuscript intended for common use will be made available on AddGene:

pLenti_hU6-pegRNA(ATP1A1-T804N)_mU6-BsmBI_Puro: ID to be provided

pLenti_PEmax-2A-BSD: ID to be provided pLenti_PEmax-2A-MLH1dn-2A-BSD: ID to be provided pLenti_PE7-2A-BSD: ID to be provided

Additional constructs are available from the lead contact upon request.

References for Response-to-Reviewers

1. Sánchez-Rivera, F.J., Diaz, B.J., Kasthuber, E.R., Schmidt, H., Katti, A., Kennedy, M., Tem, V., Ho, Y.-J., Leibold, J., Paffenholz, S.V., et al. (2022). Base editing sensor libraries for high-throughput engineering and functional analysis of cancer-associated single nucleotide variants. *Nat. Biotechnol.* 40, 862–873.
2. Kim, H.K., Yu, G., Park, J., Min, S., Lee, S., Yoon, S., and Kim, H.H. (2021). Predicting the efficiency of prime editing guide RNAs in human cells. *Nat. Biotechnol.* 39, 198–206.
3. Levesque, S., Mayorga, D., Fiset, J.-P., Goupil, C., Durringer, A., Loiselle, A., Bouchard, E., Agudelo, D., and Doyon, Y. (2022). Marker-free co-selection for successive rounds of prime editing in human cells. *Nat. Commun.* 13, 5909.
4. Mathis, N., Allam, A., Kissling, L., Marquart, K.F., Schmidheini, L., Solari, C., Balázs, Z., Krauthammer, M., and Schwank, G. (2023). Predicting prime editing efficiency and product purity by deep learning. *Nat. Biotechnol.* 41, 1151–1159.
5. Yu, G., Kim, H.K., Park, J., Kwak, H., Cheong, Y., Kim, D., Kim, J., Kim, J., and Kim, H.H. (2023). Prediction of efficiencies for diverse prime editing systems in multiple cell types. *Cell* 186, 2256–2272.e23.
6. Gould, S.I., Wuest, A.N., Dong, K., Johnson, G.A., Hsu, A., Narendra, V.K., Atwa, O., Levine, S.S., Liu, D.R., and Sánchez Rivera, F.J. (2024). High-throughput evaluation of genetic variants with prime editing sensor libraries. *Nat. Biotechnol.* <https://doi.org/10.1038/s41587-024-02172-9>.
7. Yan, J., Oyler-Castrillo, P., Ravisankar, P., Ward, C.C., Levesque, S., Jing, Y., Simpson, D., Zhao, A., Li, H., Yan, W., et al. (2024). Improving prime editing with an endogenous small RNA-binding protein. *Nature.* <https://doi.org/10.1038/s41586-024-07259-6>.
8. Halldorsson, B.V., Eggertsson, H.P., Moore, K.H.S., Hauswedell, H., Eiriksson, O., Ulfarsson, M.O., Palsson, G., Hardarson, M.T., Oddsson, A., Jensson, B.O., et al. (2022). The sequences of 150,119 genomes in the UK Biobank. *Nature* 607, 732–740.
9. Smith, M.J., Wallace, A.J., Bowers, N.L., Eaton, H., and Evans, D.G.R. (2014). SMARCB1 mutations in schwannomatosis and genotype correlations with rhabdoid tumors. *Cancer Genet.* 207, 373–378.
10. DepMap: The Cancer Dependency Map Project at Broad Institute <https://depmap.org/portal/depmap/>.
11. Landrith, T., Li, B., Cass, A.A., Conner, B.R., LaDuca, H., McKenna, D.B., Maxwell, K.N., Domchek, S., Morman, N.A., Heinlen, C., et al. (2020). Splicing profile by capture RNA-seq identifies pathogenic germline variants in tumor suppressor genes. *NPJ Precis. Oncol.* 4, 4.

Referees' report, second round of review

Reviewer #1:

In this revised manuscript, the authors have performed thorough validation experiments to reassess the cellular and molecular effects of variants in both SMARCB1 and MLH1, and have addressed all the reviewers' comments, making the manuscript suitable for publication in Cell Genomics.

Reviewer #2:

The authors did an EXCELLENT job addressing all Reviewer comments and concerns. They should be applauded for such a high quality revision, both in terms of experiments, but also in the thoughtfulness and thoroughness with which they wrote the rebuttal. Congratulations — this work will indeed set new standards in the field.

Authors' response to the second round of review

Reviewer #1: In this revised manuscript, the authors have performed thorough validation experiments to reassess the cellular and molecular effects of variants in both SMARCB1 and MLH1, and have addressed all the reviewers' comments, making the manuscript suitable for publication in Cell Genomics.

We thank Reviewer #1 for their support.

Reviewer #2: The authors did an EXCELLENT job addressing all Reviewer comments and concerns. They should be applauded for such a high quality revision, both in terms of experiments, but also in the thoughtfulness and thoroughness with which they wrote the rebuttal. Congratulations — this work will indeed set new standards in the field.

We thank Reviewer #2 for their enthusiastic support.